# Union-over-Intersections: Object Detection beyond Winner-Takes-All

**Aritra Bhowmik**      **Pascal Mettes**      **Martin R. Oswald**      **Cess G. M. Snoek**
Atlas Lab, University of Amsterdam
{a.bhowmik, p.s.m.mettes, m.r.oswald, c.g.m.snoek}@uva.nl

## Abstract

This paper revisits the problem of predicting box locations in object detection architectures. Typically, each box proposal or box query aims to directly maximize the intersection-over-union score with the ground truth, followed by a winner-takes-all non-maximum suppression where only the highest scoring box in each region is retained. We observe that both steps are sub-optimal: the first involves regressing proposals to the entire ground truth, which is a difficult task even with large receptive fields, and the second neglects valuable information from boxes other than the top candidate. Instead of regressing proposals to the whole ground truth, we propose a simpler approach: regress only to the area of intersection between the proposal and the ground truth. This avoids the need for proposals to extrapolate beyond their visual scope, improving localization accuracy. Rather than adopting a winner-takes-all strategy, we take the union over the regressed intersections of all boxes in a region to generate the final box outputs. Our plug-and-play method integrates seamlessly into proposal-based, grid-based, and query-based detection architectures with minimal modifications, consistently improving object localization and instance segmentation. We demonstrate its broad applicability and versatility across various detection and segmentation tasks.

## 1 Introduction

Object detection is a long-standing challenge in computer vision, with the goal of spatially localizing and classifying objects in images by their bounding box (Burl et al., 1998; Viola & Jones, 2004; Felzenszwalb et al., 2010). Over the past decade, significant progress has been made, driven by advances in various stages of the detection pipeline. From the foundational R-CNN (Girshick et al., 2014) to Faster R-CNN (Ren et al., 2015), and paradigm-shifting YOLO architectures (Redmon et al., 2016; Ge et al., 2021; Bochkovskiy et al., 2020; Redmon & Farhadi, 2018; Huang et al., 2018), to the recent innovations by transformer networks (Zhu et al., 2021; Carion et al., 2020; Dai et al., 2021a; Nguyen et al., 2022), these developments have enhanced feature extraction and detection. Advances in loss functions, such as focal loss (Lin et al., 2017) and IOU-aware loss (Wu et al., 2020a; Zhang et al., 2021), have further addressed class imbalance and precise localization. Regardless of backbone or optimization choices, most modern methods generate a set of proposal boxes, grid-boxes, or queries. These are classified and spatially regressed, followed by ranking or non-maximum suppression to retain only the highest scoring box per location. In essence, each proposal solves the intersection-over-union alignment with the ground truth, followed by a winner-takes-all approach. We revisit this fundamental problem of object localization and candidate selection.

We identify two issues with the common setup of object detection and propose a simple solution. First, the goal of proposal-based (Ren et al., 2015; He et al., 2017), grid-based (Redmon & Farhadi, 2018; Bochkovskiy et al., 2020) and query-based (Carion et al., 2020; Zhu et al., 2021) detectors, is to learn proposal boxes that independently represent ground truth objects. As shown in Figure 1, this often becomes an ill-posed problem. During the forward pass, proposal boxes typically capture only a portion of the ground truth, leading to an extrapolation challenge when regressing for perfect alignment. Second, there are multiple proposals or queries for each ground truth object, offering complementary views. While existing methods use box voting to select the top candidate, we focus on merging information from multiple proposals, leveraging their complementary strengths for improved detection.

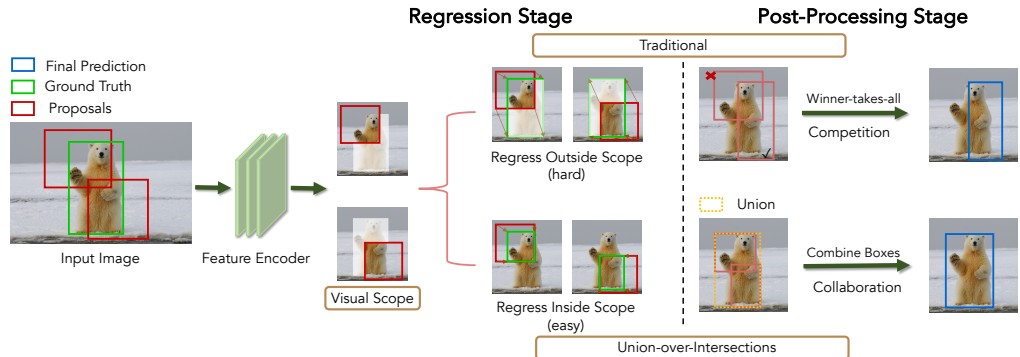

Figure 1: **Union-over-Intersections vs. Winner-takes-all.** We introduce two simple modifications to the traditional object detection pipeline. First, in the regression stage, rather than requiring proposals to align with the entire ground truth, we adjust the targets to focus solely on their intersection with the ground truth. Second, in the post-processing stage, we perform Union-over-Intersections over the traditional practice of discarding less optimal bounding boxes. Our approach underscores the advantage of cooperative interaction among proposals, demonstrating that collaboration yields superior results over competitive exclusion.

Our solution is straightforward: we decompose the problems of regressing to the entire ground truth from a single proposal and winner-takes-all candidate selection into easier to solve intersection and union problems. We do so by altering two steps of the object detection pipeline: **(i)** We set a new target for the regression of proposals: rather than predicting the entire ground truth, we only predict the region of intersection. Thus, we only regress towards the ground truth within the visual scope of the proposal. **(ii)** Given a group of proposals with predicted ground truth intersections, we form the final prediction by taking the union over the intersection regions. i.e., instead of selecting only the most confident proposal in a region, we use the wisdom of the crowd to form our final predictions.

The two stages barely affect the existing object detection pipelines. In the current regression heads, we only need to change the target coordinates. In the current non-maximum suppression, we use the exact same way of grouping proposals, but instead of a winner-takes-all, we take the union-over-intersections (**UoI**) of all proposals in a group as the final prediction. Our approach can therefore be plugged into proposal-based, grid-based, and query-based object detectors, be it a convolutional or transformer architecture. Despite its technical simplicity, UoI directly improves the detection performance. We show how our revisited approach improves canonical detection and instance segmentation methods across multiple datasets. We also demonstrate through a controlled experiment that our approach is ready to benefit from future improvements in classification and regression performance, and hope to establish it as the preferred formulation for the next generation of object detectors.

## 2 RELATED WORKS

**Object detection architectures.** Object detection architectures can be broadly categorized into: single-stage, two-stage, and transformer-based detectors, each with their own characteristics and merits. Single-stage detectors such as SSD (Liu et al., 2016), the YOLO (Redmon et al., 2016) series (including YOLOv3 (Redmon & Farhadi, 2018), YOLOv4 (Bochkovskiy et al., 2020), YOLOX (Ge et al., 2021), YOLO-Lite (Huang et al., 2018)), RetinaNet (Lin et al., 2017), EfficientDet (Tan et al., 2020), CornerNet (Law & Deng, 2018), CenterNet (Duan et al., 2019), and FCOS (Tian et al., 2019) prioritize speed, directly predicting object classes and bounding boxes in one pass. RetinaNet introduces Focal Loss for class imbalance and small object detection, while EfficientDet leverages EfficientNet (Tan & Le, 2019) backbones for scalable efficiency. Innovations in CornerNet, CenterNet, and FCOS focus on anchor-free methods, further enhancing real-time efficiency.

In the realm of two-stage object detectors, the R-CNN family has played a pivotal role. This family includes R-CNN (Girshick et al., 2014), Fast R-CNN (Girshick, 2015), and Faster R-CNN (Ren et al., 2015), which introduced end-to-end training with Region Proposal Networks. Additionally, Mask R-CNN (He et al., 2017) added segmentation capabilities, Cascade R-CNN (Cai & Vasconcelos, 2018) improved accuracy via a multi-stage framework, and Libra R-CNN (Pang et al., 2019)

addressed training imbalances. Enhancements include TridentNet (Paz et al., 2021), which handles scale variation with parallel branches, and methods like Grid R-CNN (Lu et al., 2019) for precise localization and Double-Head R-CNN (Wu et al., 2020b), which differentiates between the classification and bounding box regression, further enriching the field. Transformer-based models like DETR (Carion et al., 2020), Deformable DETR (Zhu et al., 2021), TSP-FCOS (Sun et al., 2021), Swin Transformer (Liu et al., 2021), and ViT-FRCNN (Beal et al., 2020) have revolutionized object detection. DETR views detection as set prediction, Deformable DETR adds deformable attention for focused image analysis, TSP-FCOS integrates transformers with FCOS (Tian et al., 2019) for scale variance, Swin Transformer employs a hierarchical approach, and ViT-FRCNN combines Vision Transformer (Dosovitskiy et al., 2021) with Faster R-CNN (Ren et al., 2015) for feature extraction. Follow-up work include Boxer (Nguyen et al., 2022), DINO (Zhang et al., 2022), DAB-DETR (Liu et al., 2022), UP-DETR (Dai et al., 2021b), Dynamic DETR (Dai et al., 2021a), and Conditional DETR (Meng et al., 2021), enhancing various aspects like training efficiency and dynamic predictions.

Despite the diversity in approaches, a common aspect across these three paradigms is the presence of a classification and bounding box regression stage, in which each proposal needs to individually align with the entire ground truth object. We show that proposal-based, grid-based, and query-based methods benefit from our approach of first regressing to intersections, followed by a union-over-intersection grouping.

**Object detection post-processing.** After the regression stage, selection of the best regressed proposal from a set of candidate detections ensures accuracy, with Non-Maximum Suppression (Girshick, 2015; Ren et al., 2015; Redmon et al., 2016; Liu et al., 2016) (NMS) being a key technique for selecting the best proposals by eliminating redundant detections. Traditional NMS, while effective, has been surpassed by variants such as Soft-NMS (Bodla et al., 2017), which adjusts confidence scores instead of discarding detections, improving performance in crowded scenes. Learning NMS (Hosang et al., 2017) introduces adaptability by incorporating suppression criteria into the network training. Further refinements like IoU-aware NMS (Wu et al., 2020a; Zhang et al., 2021) and Distance-IoU (DIoU) NMS (Zheng et al., 2020) consider both overlap and spatial relationships between boxes for more precise detection. Additional post-processing techniques include Matrix NMS (Wang et al., 2020), which employs a matrix operation for efficient suppression, and Adaptive NMS (Liu et al., 2019), which dynamically adjusts thresholds based on object densities. Furthermore, ASAP-NMS (Tripathi et al., 2020) optimizes processing speed, while Fitness NMS (Tychsen-Smith & Petersson, 2018) considers both detection scores and spatial fitness for more accurate suppression to enrich post-processing in object detection. Bounding box regression losses, crucial for refining predicted bounding boxes, have evolved to better align predictions with ground truth boxes. IoU loss (Yu et al., 2016; Jiang et al., 2018) directly optimizes the overlap between predicted and ground truth boxes, while GIoU loss (Rezatofighi et al., 2019) addresses scenarios where predicted boxes fall outside the ground truth, penalizing such deviations. DIoU (Zheng et al., 2020) and $\alpha$-IoU (He et al., 2021) losses further enhance regression quality by incorporating factors like the distance between box centers and aspect ratio consistency. These advanced losses have become benchmarks for evaluating the quality of bounding box regressions and are integral to improving detection accuracy in modern object detection pipelines. The improved post-processing techniques address the limitation of selecting a single winner from each cluster of proposals at a location. However, all techniques still assume that each individual box is already fully aligned with the corresponding ground truth box and serve primarily to remove duplicate detections. Instead, we seek to obtain wisdom from the crowd by using all proposals for the same object instance to form the final object detection output. We do so by making minimal adjustments to the non-maximum suppression pipeline.

## 3 Union-over-Intersections in Object Detection

Our Union-over-Intersections (UoI) approach improves the object detection pipeline with minimal adjustments, without adding new methods or architectures. We first outline the standard pipeline, focusing on bounding box regression and top candidate selection, followed by two simple modifications, as shown in Figure 2.

**Problem statement.** In traditional object detection, bounding box regression learns a mapping function $f$ that transforms a proposal box $P=(P_{x_1}, P_{y_1}, P_{x_2}, P_{y_2})$ to approximate a ground truth box $G=(G_{x_1}, G_{y_1}, G_{x_2}, G_{y_2})$. This is typically achieved by minimizing the $L1$ or $L2$ loss between the

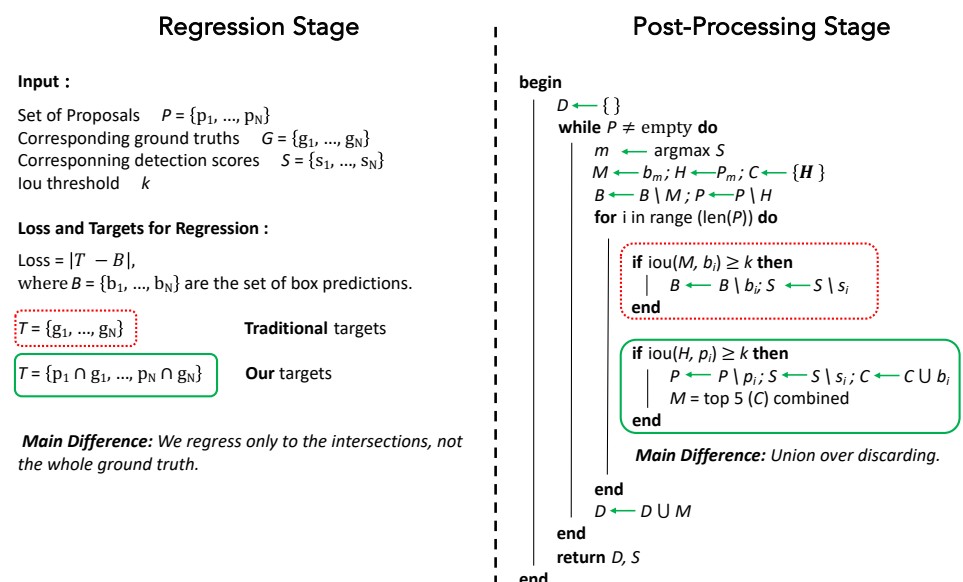

Figure 2: **Pseudo code demonstrating our minimal changes in the object detection pipeline.** During regression, we adjust the target of the proposals from the entire ground truth to only the intersection with ground truth. In post-processing, we group boxes by proposal rather than regressed outcomes and merge regressed intersections, avoiding the discard of non-maximum boxes.

predicted box and the ground truth. However, this approach presents a challenge: proposals $P$ often capture only part of the ground truth $G$, requiring $f$ to extend $P$ beyond its visible scope. As $f$ must infer parts of $G$ not visible within $P$, this becomes a difficult task. While large receptive fields help, refining proposals through a union-based approach yields better results.

Additionally, the set of proposals for a ground truth object, $\mathcal{P}=\{P_1, P_2, \ldots, P_n\}$, provides complementary views, but non-maximum suppression or ranking selects only one candidate, discarding others. This assumes the intersection-over-union problem is already solved, merely excluding duplicates in the same region.

To address these issues, we redefine box regression as an intersection learning task, aligning proposals with the intersecting parts of the ground truth boxes. Instead of discarding non-top proposals, we replace non-maximum suppression with a union operator, defining the final object as union-over-intersection of all proposals in a region. Figure 1 illustrates how these tasks lead to final object detection.

**Intersection-based regression.** We redefine the regression task as an intersection learning problem. Instead of regressing to the entire ground truth, each proposal regresses only to the visible part of the ground truth, i.e., the intersection between the proposal and ground truth. Let $I=(I_{x_1}, I_{y_1}, I_{x_2}, I_{y_2})$ be the intersection of boxes $P$ and $G$. The task is to learn a mapping function $f'$ such that $f'(P) \approx I$. The intersection is computed as:

$$I_{x_1} = \max(P_{x_1}, G_{x_1}) \ , \qquad\qquad I_{y_1} = \max(P_{y_1}, G_{y_1}) \ , \qquad (1)$$
$$I_{x_2} = \min(P_{x_2}, G_{x_2}) \ , \qquad\qquad I_{y_2} = \min(P_{y_2}, G_{y_2}) \ . \qquad (2)$$

The loss for this task is:

$$L_{\text{intersection}} = \sum_{i \in \{x_1, y_1, x_2, y_2\}} \left| f'(P_i) - I_i \right|^t \ , \qquad (3)$$

where $t$ can be 1 or 2 depending on whether the $L_1$ or $L_2$ loss is used.

**Intersection-based grouping.** Let $\mathcal{P}=\{P_1, P_2, \ldots, P_n\}$ be the set of all proposals for a ground truth object. The corresponding intersections with the ground truth are $\mathcal{I}=\{I_1, I_2, \ldots, I_n\}$. Our goal is to combine these intersections into a bounding box using a function $c : \mathcal{I} \rightarrow \mathbb{R}^4$, minimizing the

difference between the combined box and the ground truth:

$$L = \sum_{i \in \{x_1, y_1, x_2, y_2\}} |c_i(\mathcal{I}) - G_i| \ . \tag{4}$$

During inference, we refine the union of intersections to ensure a tight fit. Unlike traditional NMS, which groups regressed boxes, our method groups proposals, using the highest scoring proposal as a seed. This approach leverages the original proposals, as intersections often lack sufficient IoU for effective grouping, increasing the likelihood of fully encompassing the ground truth. Instead of discarding non-maxima, we take the minimum $[x_1^i, y_1^i]$ and maximum $[x_2^i, y_2^i]$ of all intersection regressions in the group. This forms a set of combined boxes $\mathcal{B} = \{B_1, B_2, \ldots, B_m\}$. Each combined box $B_j$ is a candidate bounding box for the object represented by group $g_j$. By design, our method yields the same number of predictions as current detectors and is compatible with any NMS variant.

Finally, we perform a regression step to refine the union-over-intersection box predictions. The regression function $r : \mathcal{B} \to \mathcal{R}$ maps each combined box to a final regressed box, denoted as $\mathcal{R}$, which represents the refined predictions:

$$L_{\text{refinement}} = \sum_{j=1}^{m} \sum_{i \in \{x, y, w, h\}} |r(B_{ji}) - T_{ji}| \ , \tag{5}$$

where $B_{ji}$ is the $i$-th coordinate of the combined box $B_j$, and $T_{ji}$ is the corresponding ground truth coordinate.

The overall loss function is a combination of the intersection loss and refinement loss:

$$L_{\text{total}} = L_{\text{intersection}} + \lambda \cdot L_{\text{refinement}} \ ,$$

where $\lambda$, set to 0.5 in our experiments, controls the weight of the refinement loss.

## 4 Experiments

### 4.1 Datasets, Evaluation, and Baseline Detector Implementations

**Datasets.** We conduct evaluations on COCO (Lin et al., 2014), covering two tasks: object detection and instance segmentation. The **MS-COCO 2017** dataset is a key benchmark for object detection and instance segmentation, comprising 80 categories with 118k training and 5k evaluation images. It features a maximum of 93 object instances per image, with an average of 7 objects. Further results on PASCAL VOC (Everingham et al., 2010) are provided in the supplemental. For **Pascal VOC**, we leverage the 2007 and 2012 editions. It spans 20 object categories, with 5,011 training and 4,952 testing images in VOC2007, and an additional 11,540 training images in VOC2012.

**Evaluation criteria.** We adhere to the standard evaluation protocols. For MS-COCO (Lin et al., 2014), we report the mean Average Precision (mAP) across IoU thresholds from 0.5 to 0.95 mAP, at specific IoU thresholds (0.5 and 0.75), multiple scales (small, medium and large), and Average Recall (AR) metrics. For PASCAL VOC (Everingham et al., 2010), we report the mAP metric at IoU thresholds of 0.5, 0.6, 0.7, 0.8 and 0.9. For instance segmentation, we report our results on MS-COCO with the mAP across IoU thresholds from 0.5 to 0.95 and across multiple scales (small, medium, large).

**Baseline detector implementations.** We evaluate our Union-over-Intersections (UoI) approach across five diverse object detection architectures: **Faster R-CNN** (Ren et al., 2015), **Mask R-CNN** (He et al., 2017), **Cascade R-CNN** (Cai & Vasconcelos, 2018), **YOLOv3** (Redmon & Farhadi, 2018), and **Deformable DETR** (Zhu et al., 2021), representing single-stage, two-stage, and transformer-based detectors. This comprehensive selection highlights the versatility of our method across different detection paradigms.

We trained Faster R-CNN, Mask R-CNN, and Cascade R-CNN on COCO using standard configurations such as random horizontal flip, 512 proposals, and SGD optimization over 12 epochs, with a learning rate decay at epochs 8 and 11. YOLOv3 is trained with the Darknet-53 backbone for 273 epochs using common augmentation strategies and SGD. Deformable DETR is trained for 50 epochs with the AdamW optimizer.

Table 1: **Detection comparison on MS-COCO.** No matter the object detection method and backbone tested, when we replace the traditional box proposal selection process with our Union-over-Intersections (***UoI***), it results in improved accuracy for all metrics.

| Method | Backbone | mAP↑ | AP$_{50}$↑ | AP$_{75}$↑ | AP$_S$↑ | AP$_M$↑ | AP$_L$↑ |
|---|---|---|---|---|---|---|---|
| Faster R-CNN (Ren et al., 2015) | R-50-fpn | 37.4 | 58.1 | 40.4 | 21.2 | 41.0 | 48.1 |
| Faster R-CNN w/ ***UoI*** | R-50-fpn | **38.1** | **58.7** | **40.9** | **21.7** | **41.8** | **49.5** |
| Faster R-CNN (Ren et al., 2015) | R-101-fpn | 39.4 | 60.1 | 43.1 | 22.4 | 43.7 | 51.1 |
| Faster R-CNN w/ ***UoI*** | R-101-fpn | **40.3** | **60.8** | **43.5** | **23.1** | **44.5** | **52.8** |
| Mask R-CNN (He et al., 2017) | R-50-fpn | 38.2 | 58.8 | 41.4 | 21.9 | 40.9 | 49.5 |
| Mask R-CNN w/ ***UoI*** | R-50-fpn | **38.8** | **59.6** | **41.8** | **22.2** | **41.6** | **50.9** |
| Mask R-CNN (He et al., 2017) | R-101-fpn | 39.8 | 60.3 | 43.4 | 23.1 | 43.8 | 52.5 |
| Mask R-CNN w/ ***UoI*** | R-101-fpn | **40.9** | **61.1** | **44.0** | **23.5** | **44.7** | **54.6** |
| Cascade R-CNN (Cai & Vasconcelos, 2018) | R-101-fpn | 42.5 | 60.7 | 46.4 | 23.5 | 46.5 | 56.4 |
| Cascade R-CNN w/ ***UoI*** | R-101-fpn | **43.1** | **61.9** | **46.8** | **24.0** | **47.2** | **57.3** |
| YOLOv3 (Redmon & Farhadi, 2018) | DarkNet-53 | 33.7 | 56.6 | 35.3 | 19.4 | 36.8 | 44.3 |
| YOLOv3 w/ ***UoI*** | DarkNet-53 | **34.5** | **57.5** | **35.9** | **19.9** | **37.3** | **45.2** |
| Def-DETR (Zhu et al., 2021) | R-50-fpn | 44.3 | 63.2 | 48.6 | 26.8 | 47.7 | 58.8 |
| Def-DETR w/ ***UoI*** | R-50-fpn | **44.8** | **63.9** | **49.1** | **27.2** | **48.3** | **59.8** |

**Inference cost.** Our method adds minimal overhead. On Faster R-CNN, we achieve 14.1 fps (compared to 14.4 fps baseline) due to the extra regression stage. In Deformable DETR, switching to part-based regression leads to slightly longer training times: 23h for 50 epochs vs. 21h for baseline.

## 4.2 BENEFIT OF UNION-OVER-INTERSECTIONS

To showcase the effectiveness of Union-over-Intersections in object detection pipelines, we perform a series of one-to-one comparisons. For Faster R-CNN, Mask R-CNN and Cascade R-CNN, we modify the regression targets to focus on object parts. For YOLO, we modify its object-to-grid assignment strategy by using an IoU-based criterion instead of the traditional center-based approach, assigning objects to multiple grid cells if they overlap sufficiently. Each anchor is tasked with regressing the part of the object it overlaps with best. In Deformable DETR, each query predicts a bounding box through the regression head after the decoder. Instead of matching these boxes to full ground truth boxes, we divide the ground truth into quadrants, assign queries to specific parts for part-based regression. Finally we apply our grouping and merging to all methods. While our method integrates with most architectures, adapting it to keypoint-based methods like FCOS requires more modifications, as they lack bounding box regions. Code is provided at `https://github.com/aritrabhowmik/UoI`.

**Detection comparison on MS-COCO.** Table 1 presents the results of our Union-over-Intersections approach on MS-COCO (Lin et al., 2014), demonstrating consistent improvements across Faster R-CNN, Mask R-CNN, Cascade R-CNN, YOLOv3 and Deformable DETR. Our method enhances detection performance across ResNet-50 and ResNet-101 backbones, as well as different architectures, highlighting its broad applicability.

In particular, we integrated our approach into **Cascade R-CNN**, which refines object proposals over multiple stages with increasing IoU thresholds. Unlike Cascade R-CNN's stage-by-stage refinement of individual proposals, our method refines the combined boxes after grouping within a single stage. We applied our intersection-based regression and union-over-intersection grouping in Cascade R-CNN's final stage, improving performance without altering the earlier stages. This demonstrates how our method can complement even complex, multi-stage architectures.

Moreover, our approach adapts seamlessly to different detection paradigms: working with proposals in two-stage detectors like Faster R-CNN, grid cells in one-stage detectors like YOLOv3, and queries in transformer-based detectors like Deformable DETR, with minimal modifications. This flexibility allows our method to enhance performance across various architectures with minimal changes.

Table 2: **Integration of IoU-based losses with UoI.** Replacing the $L_1$ loss with IoU-based losses such as GIoU, DIoU, and Alpha-IoU in our method for Faster R-CNN with a ResNet-50 backbone demonstrates consistent performance improvements, highlighting the flexibility and complementary nature of our approach.

| IoU Loss Type | Base (mAP↑) | Base w/ *UoI* (mAP↑) | Base (AP$_{75}$↑) | Base w/ *UoI* (AP$_{75}$↑) |
|---|---|---|---|---|
| $L_1$ | 37.4 | **38.1** | 40.4 | **40.9** |
| GIoU | 38.0 | **38.6** | 41.1 | **42.0** |
| DIoU | 38.1 | **38.8** | 41.1 | **41.9** |
| Alpha-IoU | 38.9 | **39.4** | 41.8 | **42.6** |

Table 3: **Instance segmentation comparison on MS-COCO.** Our Union-over-Intersections (UoI) when applied to Mask R-CNN with ResNet50 and ResNet101 backbones improves the segmentation performance. UoI is especially effective when segmenting large objects.

| Method | Backbone | mAP↑ | AP$_S$↑ | AP$_M$↑ | AP$_L$↑ |
|---|---|---|---|---|---|
| Mask R-CNN | R-50 | 34.7 | 15.8 | 36.9 | 51.1 |
| Mask R-CNN w/ *UoI* | R-50-fpn | **35.3** | **16.2** | **37.5** | **51.7** |
| Mask R-CNN | R-101 | 36.0 | 16.7 | 39.1 | 53.0 |
| Mask R-CNN w/ *UoI* | R-101-fpn | **36.8** | **17.5** | **39.8** | **53.8** |

Table 4: **UoI is independent of the grouping method**. Comparison across NMS variants with and without UoI on Faster R-CNN using ResNet-50. Results are reported on the COCO validation set.

| NMS Type | Base (mAP↑) | Base w/ *UoI* (mAP↑) | Base (AP$_{75}$↑) | Base w/ *UoI* (AP$_{75}$↑) |
|---|---|---|---|---|
| NMS | 37.4 | **38.1** | 40.4 | **40.9** |
| Cluster-NMS | 37.6 | **38.4** | 40.4 | **41.0** |
| Soft-NMS | 38.2 | **38.8** | 40.9 | **41.7** |

**Integration of IoU-based losses with UoI.** To evaluate the flexibility of our framework, we replaced the $L_1$ regression loss in our method with IoU-based losses, including GIoU (Rezatofighi et al., 2019), DIoU (Zheng et al., 2020), and Alpha-IoU (He et al., 2021). The results, shown in Table 2, confirm that our Union-over-Intersections (UoI) approach not only complements these advanced loss functions but also enhances their performance. These findings underscore the versatility of our method, which seamlessly integrates with various loss functions to achieve consistent improvements.

**Instance segmentation comparison on MS-COCO.** We extended our Union-over-Intersections (UoI) approach to the task of instance segmentation using Mask R-CNN with ResNet50 and ResNet101 backbones (Table 3). Our method consistently improves segmentation performance across various object sizes, with notable gains in mean Average Precision (mAP). Specifically, we observe an increase of 0.7 points for ResNet50 and 0.9 points for ResNet101 backbones. These improvements can be attributed to the effective merging of multiple proposals, which leads to more precise segmentation masks. The benefits are especially evident for larger objects, where tighter and more accurate localization is crucial. This demonstrates the versatility of our method, not only in object detection but also in enhancing instance segmentation tasks across different backbone architectures.

## 4.3 ABLATION STUDIES

Next we perform a series of ablation studies to pinpoint the source of our improvements.

**UoI is independent of the grouping method.** While we use traditional NMS as the canonical grouping method in our experiments, our approach is not tied to any specific grouping strategy. To evaluate this, we conducted experiments using Cluster-NMS (Zheng et al., 2021), NMS, and Soft-NMS (Bodla et al., 2017). From Table 4 we see that across all variants, our Union-over-Intersections (UoI) approach delivers consistent improvements, demonstrating its robustness and flexibility. This highlights that our method can take advantage of advancements in grouping strategies, making it adaptable to different detection frameworks and improving performance across the board.

**Our improvements come from better localization only.** We first investigate where our improved results come from. The final mAP depends both on accurate object localization and on correctly classifying objects. In Table 5, we dissect the classification and localization performance on Faster R-CNN with and without our approach. We find that the classification accuracy remains nearly

Table 5: **Classification versus localization ablation on MS-COCO**. Our approach improves the quality of the object localization without hampering classification accuracy, highlighting that union-over-intersection (*UoI*) is more viable for object detection than winner-takes-all.

| Method | Classification Accuracy [%]↑ | Localization mIoU [%]↑ |
|---|---|---|
| Faster R-CNN | 76.4 | 53.7 |
| Faster R-CNN w/ *UoI* | **76.5** | **64.4** |

Table 6: **Localization Recall Precision (LRP) comparison for Faster R-CNN on various metrics.** Replacing the traditional box proposal selection with Union-over-Intersections (*UoI*) reduces LRP errors across all metrics.

| Method | LRP (Error)↓ | LRP$_{Loc}$↓ | LRP$_{FP}$↓ | LRP$_{FN}$↓ |
|---|---|---|---|---|
| Faster R-CNN | 67.6 | 17.2 | 24.2 | 44.3 |
| Faster R-CNN w/ *UoI* | **65.3** | **12.7** | **23.9** | **43.8** |

identical, while the localization performance obtains a clear leap in mean IoU. Hence, our changes positively affect the localization quality without hampering the classification performance.

We also assess our method's performance using the *Localization Recall Precision (LRP)* metric, introduced by Oksuz et al. (2021) in Table 6. The *LRP* metric provides a comprehensive evaluation by combining localization error, false positives, and false negatives into a single error measure:

- **LRP (Error)**: Overall error that incorporates all aspects.
- **LRP$_{Loc}$**: Specifically measures localization error, reflecting how accurately bounding boxes match ground truth objects.
- **LRP$_{FP}$**: Measures false positive errors.
- **LRP$_{FN}$**: Measures false negative errors.

Our *Union-over-Intersections (UoI)* approach delivers a notable reduction in *localization error* (LRP$_{Loc}$), reducing it from 17.2 to 12.7, demonstrating improved bounding box precision. Importantly, this improvement in localization is achieved without too many changes in false positive and false negative rates, where we observe only slight reductions in *LRP$_{FP}$* (from 24.2 to 23.9) and *LRP$_{FN}$* (from 44.3 to 43.8). The stability in false positives and false negatives suggests that the more accurate the model's class predictions become, the greater the overall improvement we can achieve with our method.

**Addressing NMS limitations without architectural change.** While methods like one-to-few label assignment (Li et al., 2023) address NMS limitations by introducing entirely new architectures, our approach provides a simple, plug-and-play solution that can be applied to existing detection frameworks without architectural changes. For example, one-to-few achieves 40.9 mAP on the COCO validation set using ResNet101, through a modification of the Faster R-CNN baseline (39.4 mAP) with additional architectural changes. In contrast, our method can be seamlessly integrated into a stronger architecture like Cascade R-CNN (Cai & Vasconcelos, 2018) with the same ResNet101 backbone, achieving 43.1 mAP. This demonstrates that our approach not only avoids the complexity of redesigning architectures but also leverages existing frameworks to achieve superior results, which one-to-few cannot.

**Robustness to poor quality proposals.** One might question whether directly regressing to the ground truth from box proposals is problematic, given the large receptive fields in modern networks. In Figure 3 (a), we ablate detection performance based on the quality of the top box proposals. When proposals have high overlap with the ground truth, both methods perform similarly. However, as proposal quality drops, standard pipelines struggle. Direct regression works well only with high-quality proposals. Our approach, by looking beyond winner-takes-all, is more robust to lower-quality proposals, improving object localization.

**Regressing to intersections is simply an easier task.** One of the key hypotheses in our work is that regressing to intersection areas simplify the task compared to regressing to the entire ground truth. As shown in Figure 3(b), this simplification results in lower regression loss during training. Notably, despite our dynamic target assignment to object parts, the training remains stable and

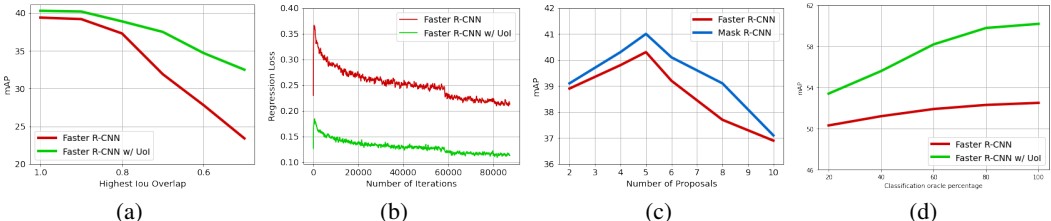

Figure 3: **Overview of four ablation studies on MS-COCO with Faster R-CNN.** (*a*) traditional object detection struggles when the best proposal has a low initial overlap with the ground truth, whereas our Union-over-Intersection, which looks beyond the winner-takes-all, is more robust to variations in proposal quality. (*b*) regressing to intersections is a simpler task to optimize as is evident from the lower loss convergence. (*c*) the optimal number of top proposals per group for the union-over-intersections is five. (*d*) as object detector classification performance improves, there are more correctly classified proposals that each cover parts of the ground truth. Instead of selecting a single proposal and discarding the rest, it is more effective to use all proposals to create a more comprehensive representation of the object.

Table 7: **Test-time augmentation benefits us further.** Both box voting and UoI involve grouping predictions but differ in approach: box voting merges regressed boxes, while UoI groups original proposals before merging intersections. Test-time augmentation enhances both methods, but UoI remains effective without it, highlighting its robustness compared to box voting's reliance on test-time augmentation. Single-scaling uses an image scale of 666x400, while Multi-scaling incorporates both 666x400 and 2000x1200 as image scales.

| | Test-time adaptation setting | | | |
|---|---|---|---|---|
| Method | None | Flip | Flip + Single-scaling | Flip + Multi-scaling |
| Faster R-CNN w/ Box voting | 37.5 | 37.7 | 38.0 | 38.8 |
| Faster R-CNN w/ *UoI* | **38.1** | **38.2** | **38.6** | **39.3** |

achieves improved localization accuracy during inference, as evidenced by Table 5. We conclude that regressing to intersections not only simplifies object detection but also enhances final detection performance through the union-over-intersections strategy.

**Intermediate group size is best for union-over-intersection.** Figure 3 (c) shows the object detection performance as a function of the number of proposals used per group when taking the union over intersections. Our approach consistently uses a maximum of five top proposals per region across all datasets and architectures, with all available proposals used if fewer than five are present. Performance improves from 2 to 5 proposals, then drops due to noisy, low-scoring proposals, achieving the best results with 5 proposals. This setting balances sufficient information for box merging while avoiding overly large boxes.

**Test-time augmentation benefits us further.** We analyze the impact of test-time augmentation (TTA) on our Union-over-Intersections (UoI) strategy and compare it with box voting (Roman et al., 2019), which involves confidence-weighted box merging. TTA involves applying multi-scale testing and horizontal flips during inference, generating additional predictions for aggregation. Box voting relies heavily on the increased diversity of predictions from TTA, while UoI proves effective even without TTA and further benefits when combined with it.

**With better classification, UoI becomes more powerful.** We examine whether improvements in object classification will reduce or expand the gap between traditional methods and our approach. In an oracle experiment, we simulate varying proportions of known ground truth labels (20%, 40%, 50%, 60%, 80%, and 100%) while the remaining predictions come from the network. As shown in Figure 3 (d), as the accuracy of classification labels improves, our Union-over-Intersections method increasingly outperforms traditional techniques. This is because correctly classified proposals, each covering parts of the ground truth, combine to form a more comprehensive representation. In contrast, traditional methods, which rely on a single proposal, often miss portions of the object. These results highlight that as classification improves, the advantages of UoI become more pronounced.

Table 8: **Effectiveness of the second regression stage.** UoI inherently involves a second regression stage, where the combined intersections are further refined for improved performance. Adding just a second regression stage to the Faster R-CNN baseline alone does not yield similar benefits.

| Method | mAP↑ | AP$_{75}$↑ |
|---|---|---|
| Faster R-CNN | 37.4 | 40.4 |
| Faster R-CNN w/ 2nd regression | 37.4 | 40.5 |
| Faster R-CNN w/ *UoI* | **38.1** | **40.9** |

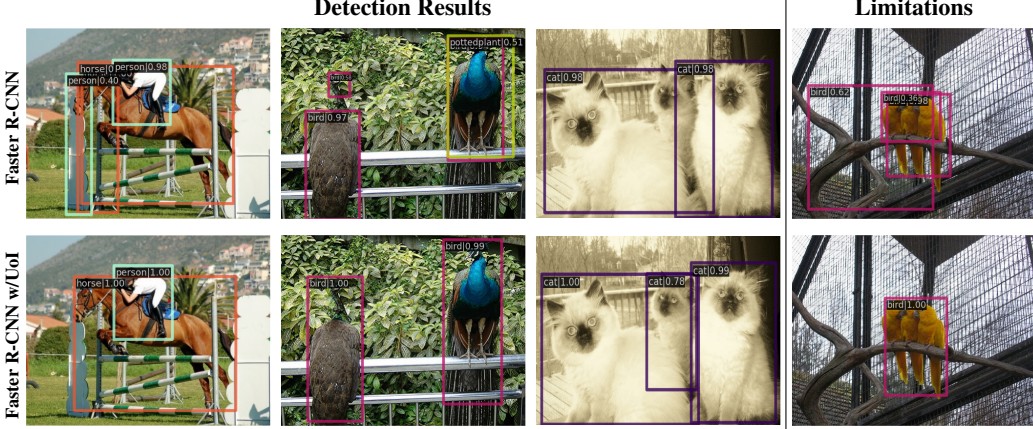

Figure 4: **Qualitative analysis** showcasing the effectiveness of our method applied to Faster R-CNN for object detection. Starting from the left, our method successfully removes the incorrect 'person' prediction, precisely localizes the entire bird, and identifies the third cat as a distinct entity. However, in scenarios with clutter or a lot of overlap, as observed in the right image, our approach also consolidates multiple parrots into a single detection.

**Effectiveness of the second regression stage.** We include an ablation study to analyze the necessity of the second regression stage in our method. Unlike baseline methods, where an additional regression head does not provide benefits due to redundant tasks, our approach assigns distinct roles to the regression stages. The first head focuses on parts of the object, while the second refines the union of these parts into a cohesive bounding box. As shown in Table 8, this distinction makes the second regression stage effective, further improving performance.

**Limitations.** Our method benefits from multiple proposals, allowing the Union-over-Intersection strategy to unify their information effectively. However, for small objects, if only one proposal is available, our approach defaults to the baseline. Figure 4 highlights another limitation with crowded same-class objects, where closely positioned instances may merge into a single box ('3 birds' example). Advanced grouping strategies could mitigate this issue.

## 5 CONCLUSION

In this paper, we introduced a plug-and-play approach to object detection that enhances localization by focusing on the intersection between proposals and ground truth and applying union-over-intersections instead of a winner-takes-all strategy. These simple changes, requiring minimal modifications to existing architectures, consistently improve performance across two-stage, single-stage, and transformer-based detectors for object detection as well as instance segmentation. Our method proves especially robust for stricter overlap thresholds and lower-quality proposals, making it highly flexible across various frameworks. By improving localization, our approach offers a practical solution that strengthens object detection and segmentation without architectural dependencies.

**Acknowledgement.** This work has been financially supported by TomTom, the University of Amsterdam and the allowance of Top consortia for Knowledge and Innovation (TKIs) from the Netherlands Ministry of Economic Affairs and Climate Policy. We also extend our gratitude to the anonymous reviewers for their valuable feedback and insightful suggestions during the rebuttal stage, which considerably improved this work.

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

# A APPENDIX

## A.1 ADDITIONAL RESULTS ON PASCAL VOC

To provide a more comprehensive evaluation, we include results for Faster R-CNN, Mask R-CNN, Cascade R-CNN, YOLOv3 and Deformable DETR on the PASCAL VOC dataset. As shown in Table 9, our Union-over-Intersections (UoI) strategy also consistently improves detection performance across all methods on PASCAL VOC, with again stronger gains observed at higher overlap thresholds.

Table 9: **Comparison on PASCAL VOC with additional methods.** Adding UoI improves performance across all methods, particularly for higher overlap thresholds.

| Method | Backbone | $AP_{50}\uparrow$ | $AP_{70}\uparrow$ | $AP_{90}\uparrow$ |
|---|---|---|---|---|
| Faster R-CNN | R-50-fpn | 71.2 | 51.3 | 6.8 |
| Faster R-CNN w/ *UoI* | R-50-fpn | **72.9** | **55.3** | **11.3** |
| Mask R-CNN | R-50-fpn | 73.1 | 52.7 | 7.9 |
| Mask R-CNN w/ *UoI* | R-50-fpn | **74.0** | **56.9** | **13.1** |
| Cascade R-CNN | R-50-fpn | 77.5 | 54.1 | 10.5 |
| Cascade R-CNN w/ *UoI* | R-50-fpn | **78.8** | **57.9** | **14.8** |
| YOLOv3 | DarNet-53 | 56.1 | 41.7 | 5.3 |
| YOLOv3 w/ *UoI* | DarNet-53 | **57.0** | **44.2** | **8.8** |
| Deformable DETR | R-50-fpn | 83.2 | 68.5 | 22.1 |
| Deformable DETR w/ *UoI* | R-50-fpn | **84.1** | **70.3** | **26.5** |

## A.2 EFFECT OF IOU THRESHOLD ON GROUPING STRATEGY

To evaluate the impact of the IoU threshold ($k$) on our UoI strategy, we conducted an ablation study using Faster R-CNN with a ResNet-101 backbone. The IoU threshold influences group sizes during the grouping stage: lowering the threshold (e.g., 0.3) increases group sizes, while raising it (e.g., 0.7) decreases group sizes. As shown in Table 10, the optimal number of proposals shifts to 4 for $k=0.7$ but remains at 5 for $k=0.3$. Importantly, the highest mAP is achieved with $k=0.5$, reaffirming the effectiveness of the default setting. These results further validate the robustness of our approach to the IoU threshold choice.

Table 10: **Effect of IoU threshold ($k$) on grouping strategy.** Lowering or raising the IoU threshold shifts the optimal number of proposals, but the highest mAP is consistently achieved with $k = 0.5$.

| IoU Threshold $k$ | Number of Proposals | | | | |
|---|---|---|---|---|---|
| | 3 | 4 | 5 | 6 | 7 |
| 0.3 | 38.1 | 39.0 | 39.5 | 39.2 | 38.4 |
| 0.5 | 39.3 | 39.8 | 40.3 | 39.2 | 38.5 |
| 0.7 | 38.5 | 39.9 | 39.4 | 38.1 | 37.7 |

## A.3 IMPACT OF ADDITIONAL CLASSIFICATION HEAD ON CONFIDENCE SCORES

To assess the effect of adding a classification head for the combined boxes, we conducted an ablation using the Faster R-CNN baseline. As shown in Table 11, this modification resulted in marginal improvements in $mAP$ and $AP_{75}$. However, since the gains are minimal and the primary improvements in our approach stem from better localization (see Table 5 in the main paper), we did not include the additional classification stage in our main method. These results are provided here for completeness.

## A.4 EVALUATING FULL-OBJECT REGRESSION VERSUS PART-BASED REGRESSION

To evaluate the implications of applying the Union of Proposals with the original regression objective, we conducted an experiment using full-object regression, where the model predicts the complete ground truth bounding boxes instead of focusing on part-based regions. As shown in the table

Table 11: **Effect of adding a classification head for combined boxes.** Adding a classification stage offers minimal improvements, reaffirming that the primary gains of our approach come from improved localization

| Method | mAP↑ | AP$_{75}$↑ |
|---|---|---|
| Faster R-CNN w/ *UoI* | 38.1 | 40.9 |
| Faster R-CNN w/ *UoI* and 2nd classification stage | **38.2** | **41.1** |

below, full-object regression results in looser bounding boxes after combining proposals, even with a second-stage regression. This leads to a considerable drop in performance compared to our part-based regression strategy (UoI). The results confirm that focusing on parts of objects enables finer localization and better overall accuracy.

Table 12: Comparison of full-object regression versus part-based regression (UoI). Part-based regression achieves better performance by enabling finer localization.

| Method | mAP↑ | AP$_{75}$↑ |
|---|---|---|
| Full-object regression | 35.8 | 37.5 |
| Part-based regression (*UoI*) | **38.1** | **40.9** |

## A.5 ADAPTATION OF UoI FOR DEFORMABLE DETR AND YOLO

Here we provide more details on how our approach integrates with Deformable DETR and YOLO. Code is provided at https://github.com/aritrabhowmik/UoI.

In the standard Deformable DETR, each query predicts a bounding box via the regression head after the decoder. These bounding boxes are then matched to full ground truth boxes using Hungarian matching, with the best query assigned to each ground truth box. Our method modifies this matching process. Specifically, we divide each ground truth box into four quadrants (top-left, top-right, bottom-left, bottom-right), treating these quadrants as part-level targets. During training, each query is assigned to a specific quadrant based on its IoU overlap, and the regression head is tasked with predicting the corresponding part of the ground truth. At inference, these part-level predictions are grouped using our Intersection-based Grouping strategy and merged into complete bounding boxes.

Similarly, for YOLO, we adapt its object-to-grid assignment strategy. Instead of assigning an object to a single grid cell based on the object's center, we use an IoU-based criterion. An object is assigned to multiple grid cells if it overlaps sufficiently with their anchors. Each anchor predicts the part of the object it best overlaps with, and during inference, our grouping strategy merges these predictions into final bounding boxes. These modifications retain the original architecture of YOLO while enabling part-based regression and grouping.

## A.6 MORE QUALITATIVE RESULTS

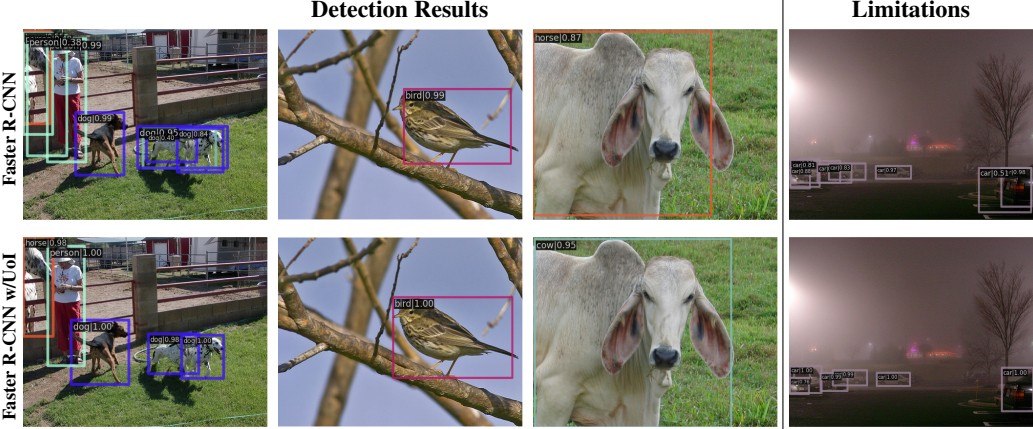

Figure 5: **Qualitative analysis** showcasing the effectiveness of our method applied to Faster R-CNN for object detection. Starting from the left, our UoI approach diminishes false positives and achieves tighter localization for both the bird and the cow, outperforming the baseline method in detecting large objects. However, in scenarios with clutter or significant overlap, our method may also aggregate objects into a single group, as seen in the rightmost image with the two cars grouped as one in the right corner.

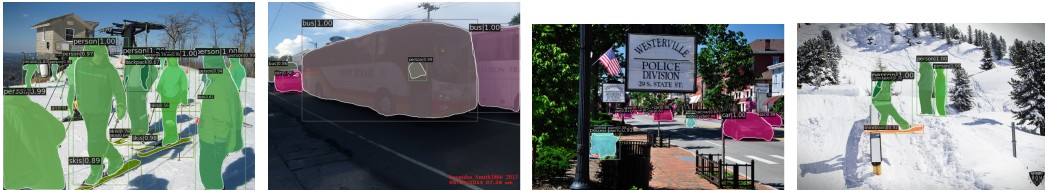

Figure 6: **Qualitative analysis** of our method for instance segmentation on the COCO 2017 dataset. Our method merges information from multiple proposals, naturally leading to precise localization and segmentation masks.

