# OpenReview forum: "Union-over-Intersections: Object Detection beyond Winner-Takes-All"
_ICLR.cc/2025/Conference — ICLR 2025 Spotlight_

### Official Review · Reviewer_JFjT · 2024-10-31

**Soundness:** 3
**Presentation:** 3
**Contribution:** 3
**Rating:** 8
**Confidence:** 5

**Summary:**

The paper proposes to predict a set of intersection boxes for one given ground-truth box and then take the union of them to form the final detection box. The method is simple yet effective and is verified on 5 classic detectors.

**Strengths:**

1. The proposed method is simple yet effective and easy to be re-produced.

2. The discussion of the problem is quite clear; the paper is well-written.

3. The experiments and analyses are sufficient to support the soundness of the proposed method over the vanilla regression method. The results also verify its good versatility for a variety of object detectors.

**Weaknesses:**

1. As a box regression method, the paper did not compare the proposed method with IoU-based loss functions, e.g., GIoU [r1], CIoU [r2], Alpha-IoU [r3], EIoU [r4], etc, making the superiority of the methods less convincing.

2. It is mentioned by line 216, the proposed Intersection-based Grouping is compatible with any NMS variant. In the experiment, only Soft-NMS is tested. I suggest the authors provide more ablation on this.

3. In Fig.3(c), it shows that the performance may be sensitive to the number of proposals on Faster R-CNN and MS COCO. This characteristic may limit its robustness in different scenarios, e.g., different detectors or different datasets.

4. Why the datasets of ablation studies switch frequently between COCO and VOC? Not consistent.

5. The font size of Fig.3 is too small.

[r1] Rezatofighi H, Tsoi N, Gwak J Y, et al. Generalized intersection over union: A metric and a loss for bounding box regression. CVPR 2019.

[r2] Zheng Z, Wang P, Liu W, et al. Distance-IoU loss: Faster and better learning for bounding box regression. AAAI 2020.

[r3] He J, Erfani S, Ma X, et al. $\alpha $-IoU: A family of power intersection over union losses for bounding box regression. NeurIPS 2021.

[r4] Zhang Y F, Ren W, Zhang Z, et al. Focal and efficient IOU loss for accurate bounding box regression[J]. Neurocomputing, 2022.

**Questions:**

Does the difference between the middle group of Table 1 (Faster R-CNN) and Table 4 lie in whether Soft-NMS is applied or not?

---

> ### Author Response · Authors · 2024-11-24
>
> We thank the reviewer for their appreciation and the actionable feedback to improve our experimental rigor.
>
> > **IoU-based loss functions**
>
> We clarify our method is complementary to existing IoU-based loss functions, not a competitor. To address the reviewer's comment, we performed additional experiments combining our method with the publicly available implementations of GIoU [a], DIoU [b], and Alpha-IoU [c] . Results are presented in the table below, using a ResNet-50 backbone on the COCO-val dataset. Integrating these losses into our approach is straightforward and just requires replacing the L1 loss with the respective IoU-based loss during the regression stage.The results confirm consistent improvements, demonstrating that our approach not only works with these advances but also enhances their performance. We will include this experiment and discussion in the supplemental material.
>
> | IoU Loss Type | Base ($mAP$) | Base w/ UoI ($mAP$) | Base ($AP_{75}$) | Base w/ UoI ($AP_{75}$) |
> |---------------|------------|--------------------|--------------|----------------------|
> | GIoU            | 38.0       | **38.6**   | 41.1         | **42.0**     |
> | DIoU            | 38.1       | **38.8**   | 41.1         | **41.9**     |
> | Alpha-IoU    | 38.9       | **39.4**   | 41.8         | **42.6**     |
>
> [a] Rezatofighi H, et al. Generalized intersection over union: A metric and a loss for bounding box regression. CVPR 2019.
>
> [b] Zheng Z, et al. Distance-IoU loss: Faster and better learning for bounding box regression. AAAI 2020.
>
> [c] He J, et al. $\alpha $-IoU: A family of power intersection over union losses for bounding box regression. NeurIPS 2021.
>
> ___
>
> > **Effect of different nms variants**
>
> Our method is indeed compatible with various NMS variants, as they primarily influence the grouping stage. Following the reviewer suggestion, we conducted experiments with Cluster-NMS [d], comparing it to NMS and Soft-NMS [e]. The results confirm consistent improvements across all variants, demonstrating the robustness of our approach. We will include this ablation in the supplemental material.
>
> | NMS Type | Base ($mAP$) | Base w/ UoI ($mAP$) | Base ($AP_{75}$) | Base w/ UoI ($AP_{75}$) |
> |---------------|------------|--------------------|--------------|----------------------|
> | NMS             | 37.4       | **38.1**   | 40.4         | **40.9**     |
> | Cluster-NMS]            | 37.6       | **38.4**   | 40.4         | **41.0**     |
> | Soft-NMS      | 38.2       | **38.8**   | 40.9         | **41.7**     |
>
> We use a Resnet50 backbone for the Faster-RCNN and the results are on the COCO val set. All variants are trained with L1 regression loss.
>
> [d] Zheng Z et al. Enhancing Geometric Factors in Model Learning and Inference for Object Detection and Instance Segmentation. IEEE transactions on cybernetics, 2021
>
> [e] Bodla N, et al. Soft-NMS--improving object detection with one line of code. ICCV 2017
>
> ___
>
> > **Number of proposals**
>
> Overall, we consider our approach robust, as we consistently use a maximum of 5 top proposals per region across all datasets and architectures. This hyperparameter ensures a balance between having sufficient information for accurate box merging and avoiding overly large boxes when too many proposals are combined. If there are fewer than 5 proposals for a ground truth target, all available proposals are used; if there are more, only the top 5 are considered. We will clarify this robustness and consistency in the paper.
>
> ___
>
> > **More consistency between datasets used for ablations**
>
> We thank the reviewer for raising this concern. To clarify, the majority of our ablation studies are conducted on the COCO dataset for consistency except for Table 4 in the main paper, where we present an ablation on Soft-NMS using the VOC dataset. To address this concern, we have now included the same ablation on the COCO dataset in this rebuttal. Moving forward, we will ensure that results for all ablation studies are presented consistently across datasets, with detailed COCO results included in the supplemental material.
>
> ___
>
> > **Font size in Figure 3**
>
> We will increase the font size in Fig. 3 to improve readability in the revised version.
>
> ___
>
> > **Soft NMS in Table 4**
>
> Indeed, the difference between Tables 1 and 4 for the Faster R-CNN backbone is due to the use of Soft NMS. We will make sure to include NMS and Cluster-NMS results in Table 4 as well to have an easy overview of this comparison.
>
> ___
>
> We hope to have addressed the concerns and demonstrated the robustness, flexibility, and broad applicability of our approach, including its compatibility with IoU-based losses, multiple NMS variants, and diverse datasets. We gladly clarify any remaining questions.

---

> > ### Comment · Reviewer_JFjT · 2024-11-25
> >
> > The authors' rebuttal addresses my concerns. However, the comparison with the IoU-based losses is so important and needs to be placed in the main paper. I suggest moving Table 1 to the supplementary material, since the PASCAL VOC dataset is a bit outdated, and Table 2 serves the same purpose and is more convincing than Table 1.

---

> > > ### Author Response · Authors · 2024-11-26
> > >
> > > We are grateful to hear that the reviewer’s concerns have been addressed. As suggested, we have moved Table 1 with PASCAL VOC results to the supplemental material and included the experiment demonstrating the complementarity of our approach to IoU-based losses in the updated main paper. For consistency, we have also replaced Table 4 with the NMS ablation on PASCAL VOC with the one on COCO. Thank you for the valuable feedback.

---

> > > > ### Author Response · Authors · 2024-11-27
> > > >
> > > > Dear reviewer,
> > > >
> > > > As the discussion stage is ending soon, we wonder if our response answers your questions and addresses your concerns? Thanks again for your very constructive and insightful feedback!

---

> ### Comment · Area_Chair_cV87 · 2024-11-25
>
> Dear Reviewer JFjT,
>
> Could you kindly review the rebuttal thoroughly and let us know whether the authors have adequately addressed the issues raised or if you have any further questions.
>
> Best,
>
> AC of Submission10907

---

> ### Comment · Reviewer_JFjT · 2024-11-29
>
> There are still important ablations in Appendix that do not appear in the main paper. The Fig. 4 is too large. I suggest removing half of Fig. 4 to save more room for presenting those experiments. Fig.5 is showing something similar to Fig.4 and can be removed too.   I argue that those ablation studies are inspiring that provide more insights of the method.
>
> The followings are minor concerns:
>
> 1) There are no related works of bbox regression in Sec.2. I suggest giving a comprehensive introduction to that.
>
> 2) All the algorithms engaging in comparison must be appears in Sec.2 Related Works. Please check them carefully.
>
> I increase my rating score to 7.

---

> > ### Author Response · Authors · 2024-12-01
> >
> > We kindly accept your suggestions to move more ablations to the main paper by updating Figures 4 and 5 into a single, smaller figure, and expanding related work in accordance with your guidance.
> >
> > Thank you for helping us improve our work and upgrading your score.

---

### Official Review · Reviewer_fZZc · 2024-11-03

**Soundness:** 3
**Presentation:** 4
**Contribution:** 2
**Rating:** 5
**Confidence:** 4

**Summary:**

The authors propose the Union-over-Intersections (UoI) method, which modifies the object detection pipeline in two key ways:

1. **Intersection-Focused Regression** : Instead of regressing proposals to the entire ground truth, the method focuses on the intersection area, simplifying the regression task and improving localization accuracy.
2. **Union of Proposals** : In the post-processing stage, rather than using a winner-takes-all strategy, the UoI method combines information from all proposals by taking the union of their regressed intersections, enhancing the final detection output.

The paper demonstrates decent improvements in object localization and instance segmentation across various architectures, including Faster R-CNN and YOLOv3. The UoI method is adaptable and can be integrated into existing detection frameworks with minimal changes, making it a promising advancement in the field of object detection.

**Strengths:**

* The method is straightforward, requiring only modifications to the training objectives and post-processing steps, making it easy to implement within existing proposal-based detectors.
* Experimental results demonstrate a solid improvement in accuracy while introducing only a minimal additional computational overhead

**Weaknesses:**

My primary concern lies in the novelty and significance of the method. The paper emphasizes the use of intersections, but in my view, intersections are a subset of Intersection-over-Union (IoU). Learning IoU inherently involves understanding intersections, and there are already existing works focused on learning IoU, such as IoU loss. This diminishes the overall importance of the paper. Additionally, learning the complete ground truth bounding box implies the need to understand both intersections and unions. With proper tuning, I believe that incorporating more information would benefit neural network learning rather than focusing solely on a partial representation.

The concept of the Union of Proposals appears to function similarly to a voting mechanism. Box voting for accuracy improvement has already been extensively validated through test-time augmentation techniques in challenges like VOC and COCO. Moreover, learning the complete ground truth bounding box can naturally facilitate a voting process as well.

Lastly, there are concerns regarding generalizability. The method relies on proposals, which limits its applicability in proposal-free approaches, such as YOLO and FCOS. This reliance on a specific number of proposals necessitates tuning, making integration with proposal-free methods less convenient.

**Questions:**

1. What would be the implications of applying the Union of Proposals to learn the complete ground truth bounding boxes without altering the learning objectives?
2. Please provide a rationale for the weaknesses summarized above.

---

> ### Author Response · Authors · 2024-11-24
> **Response 1/2**
>
> We thank the reviewer for thoughtful feedback and the opportunity for reconsideration.
>
> > **Difference between an Iou-loss regression objective and our approach**
>
> We clarify that the key novelty of our method lies in its redefinition of how object detection is approached. Unlike IoU-based methods that focus on optimizing bounding box accuracy through new loss functions, our approach shifts the focus to learning part-wise representations of an object. By fostering collaboration among these parts, we unify and reconstruct the final object, offering a fundamentally different perspective on object detection.
>
> In fact, our framework’s flexibility allows it to incorporate any regression loss function or grouping strategy and integrate seamlessly with various detection architectures. To showcase this, we conducted an additional experiment where we replaced the L1 loss in our method with IoU-based losses like GIoU [a], DIoU [b], and Alpha-IoU [c]. The results, shown in the table below, confirm consistent improvements, demonstrating that our approach not only works with these advances but also enhances their performance. We will include this experiment and discussion in the supplemental. Thank you.
>
> | IoU Loss Type | Base ($mAP$) | Base w/ UoI ($mAP$) | Base ($AP_{75}$) | Base w/ UoI ($AP_{75}$) |
> |---------------|------------|--------------------|--------------|----------------------|
> | GIoU            | 38.0       | **38.6**   | 41.1         | **42.0**     |
> | DIoU             | 38.1       | **38.8**   | 41.1         | **41.9**     |
> | Alpha-IoU     | 38.9       | **39.4**   | 41.8         | **42.6**     |
>
> [a] Rezatofighi H, et al. Generalized intersection over union: A metric and a loss for bounding box regression. CVPR 2019.
>
> [b] Zheng Z, et al. Distance-IoU loss: Faster and better learning for bounding box regression. AAAI 2020.
>
> [c] He J, et al. $\alpha $-IoU: A family of power intersection over union losses for bounding box regression. NeurIPS 2021.
>
> ___
>
> > **Box voting**
>
> We acknowledge that "box voting for accuracy improvement" is an established and extensively validated method, that is why we use it, but to group and merge proposals, instead of selecting only the top one. This is where we are different. We agree that learning the complete ground truth will facilitate in selecting the best candidate. We note that we do not select the best candidate. We merge information from the top-k best candidates, thus making our combined candidate better than the top-1 voted best candidate as done in the existing literature. We will better stress this difference in the introduction.
>
> ___
>
> > **Use beyond proposal-based methods**
>
> We clarify our method is not limited to traditional proposals and can be applied to methods with regions, grids, or queries, as demonstrated in the main paper (Tables 1 and  2). We have already shown its integration with grid-based YOLO and query-based Deformable DETR, highlighting its flexibility. The concept of "proposals" is used loosely to includede regions and anchors, which are common across most detection frameworks, making our approach broadly applicable with minimal modifications.
>
> **Details of modifications:** For YOLO, we modify its object-to-grid assignment strategy by using an IoU-based criterion instead of the traditional center-based approach, assigning objects to multiple grid cells if they overlap sufficiently. Each anchor is tasked with regressing the part of the object it overlaps with best, and during inference, our Intersection-based Grouping merges these predictions. For Deformable DETR, we treat queries as proposals and redefine the matching process to focus on object parts, assigning queries to specific regions of the ground truth and combining them during inference.
>
> While most methods, including YOLO and Deformable DETR, can incorporate our Union-over-Intersection with minor modifications, unique architectures like FCOS, which rely on keypoints rather than rectangular anchors, may require additional adaptations. We will clarify the adaptability of our approach in the main paper.
>
> ___
>
> **<CONTINUED>**

---

> ### Author Response · Authors · 2024-11-24
> **Response 2/2**
>
> > **Applying union-over-intersection to full ground truth predictions**
>
> Following the reviewer suggestion, we apply the Union of Proposals while maintaining the original regression objective (Full-object regression), where the model predicts full ground truth bounding boxes instead of part-based regions. As shown in the table below, this approach produces looser bounding boxes after combining proposals, even with a second-stage regression, resulting in lower AP scores. This confirms the necessity of our part-based regression strategy (UoI), which achieves finer localization and better overall accuracy by focusing on parts of the objects. We will include this analysis in the supplemental material.
>
> | Method                     | $mAP$    | $AP_{75}$ |
> |-----------------------------|-------|-------|
> | Full-object regression    | 35.8  | 37.5  |
> | Part-based regression (UoI) | **38.1**  | **40.9**  |
>
> ___
>
> We hope to have clarified the novelty of our method, its distinction from IoU-based approaches and traditional voting mechanisms, and its adaptability to various architectures, including proposal-free methods. We gladly clarify any remaining questions.

---

> > ### Comment · Reviewer_fZZc · 2024-11-25
> >
> > Thanks for the feedback from the authors.
> >
> > In my view, this merge complicates further voting efforts.  Therefore,  It would be better if the author could provide a quantitative comparison between the improvements from the union proposal and those from voting.
> > Table 2 in the paper reports a 0.7 improvement using UoI with R-50-FPN. However, based on my experience, voting can also achieve improvements exceeding 0.7.

---

> > > ### Author Response · Authors · 2024-11-26
> > >
> > > > **Comparison with test-time augmentation and box voting**
> > >
> > > We now realize we misunderstood your initial question and missed the test-time augmentation (TTA) effect on our method. Our new experiment on the COCO validation set with Faster R-CNN (ResNet-50-FPN) demonstrates that our approach also benefits from test-time augmentation, where we perform data augmentation (random flip and image resize) during inference, giving us an additional 0.5 mAP gain. Box voting (confidence-weighted box merging)  is especially effective with test-time augmentation, while our Union-over-Intersections is already effective without, and improves further with test-time augmentation. We have included this result in the updated main paper. Thank you for this suggestion.
> > >
> > > | Method                  | Box Voting  | UoI  |
> > > |-------------------------|--------|--------|
> > > | Faster R-CNN                    | 37.5   | 38.1   |
> > > | Faster R-CNN  + TTA        | 38.0   | 38.6   |

---

> > > > ### Author Response · Authors · 2024-11-27
> > > >
> > > > Dear reviewer,
> > > >
> > > > As the discussion stage is ending soon, we wonder if our response answers your questions and addresses your concerns? Thanks again for your very constructive and insightful feedback!

---

> > > > > ### Comment · Reviewer_fZZc · 2024-11-28
> > > > >
> > > > > Thanks for the feedback. Based on the author's results,  UoI appears to be more effective than using bounding box voting. However, the baseline in this table isn't very convincing to me. Firstly, compared to the Faster R-CNN baseline (37.4) in Table 2, the voting method only improves by 0.1, which means there's almost no gain. Additionally, using multiscale + flip + voting only increases the baseline by 0.6, which falls far short of the improvements typically observed in the COCO Challenge.
> > > > >
> > > > > Back to my question. Voting requires multiple results to perform the process, but this work's merging approach has already combined them into a single result, making further voting difficult. In other words, both voting and UoI are methods that require merging results. Although the author shows that UoI performs better than voting, the voting results are so low that it's difficult for me to assess the paper effectively.

---

> > > > > > ### Author Response · Authors · 2024-11-29
> > > > > >
> > > > > > We thank the reviewer for continued engagement and helping us making the case for our work. The reviewer is right that more improvements can be obtained, for example by optimizing multi-scale augmentation parameters. In the table below, we report the impact of such parameters on the COCO validation set. Weighted box merging improves from 37.5 to 38.8 (+1.3), and our approach improves as well from 38.1 to 39.3 (+1.2). We will include these updated results in the supplemental.
> > > > > >
> > > > > > To answer your question, in our understanding box voting based on confidence values is happening during the proposal grouping process. Unlike traditional box voting, which groups regressed boxes, our method groups the original proposals. Hence, like box voting, our approach profits from having more proposals to group. After the grouping process we merge regressed intersections, followed by a regression step, whereas box voting merges regressed bounding boxes. We will better stress these subtle similarity and differences in the main paper, and hope to have resolved your question.
> > > > > >
> > > > > > |                          |  Test-time augmentation(s)   | **Weighted Box Merging** | **UoI (ours)** |
> > > > > > |----------------------------------------|--------------------|--------------------|---------------|
> > > > > > | Faster R-CNN  | –                     | 37.5              | 38.1          |
> > > > > > | Faster R-CNN | Flip                        | 37.7              | 38.2          |
> > > > > > | Faster R-CNN | Flip + Scale(666x400)            | 38.0              | 38.6          |
> > > > > > | Faster R-CNN | Flip + Scale(666x400) + Scale(2000x1200)  | 38.8              | 39.3          |

---

> > > > > > > ### Author Response · Authors · 2024-12-02
> > > > > > >
> > > > > > > Dear reviewer,
> > > > > > >
> > > > > > > As the discussion stage is ending soon, we wonder if our response answers your questions and addresses your concerns? Thanks again for your very constructive and insightful feedback!

---

> ### Comment · Area_Chair_cV87 · 2024-11-25
>
> Dear Reviewer fZZc,
>
> Could you kindly review the rebuttal thoroughly and let us know whether the authors have adequately addressed the issues raised or if you have any further questions.
>
> Best,
>
> AC of Submission10907

---

### Official Review · Reviewer_GWg2 · 2024-11-05

**Soundness:** 4
**Presentation:** 3
**Contribution:** 3
**Rating:** 8
**Confidence:** 4

**Summary:**

This paper propose Union-over-Intersections method for object detection, which predict intersection box between proposal box and ground truth box and combine overlapped boxes into the final instance.

**Strengths:**

- The authors provide a thorough explanation of the methodology, making it easy to understand.
- The authors present a novel insight and have correctly implemented the methodology, achieving promising results.
- The experimental content is comprehensive, covering key results across popular detection algorithms and both detection and segmentation tasks. The ablation study effectively discusses the strengths of the approach and includes an analysis of its limitations.

**Weaknesses:**

- While this paper attempts to challenge the conventional box refinement paradigm (predict box directly and Winner-Takes-All for post-processing) in object detection models, it does not sufficiently demonstrate its viability as a true alternative to the original paradigm. (NOTE that although I mention this as a weakness, I still consider the approach to be novel and appreciate their effort to explore new insights. However, from an intuitive standpoint, the method does not strongly compel me to adopt it.)

**Questions:**

- Intuitively, this method seems more beneficial for detecting larger objects, as the "combine box" approach tends to union the box when merging bounding boxes within a group, potentially biasing towards larger boxes. Is combining multiple predictions still an ideal detection strategy for small objects? The smaller targets are often adequately covered by a single proposal, which means the approach sometimes defaults to the standard paradigm. Although the main experimental results indicate an improvement in mAP across object sizes, I would encourage the authors to further analyze how this paradigm performs for targets of different sizes.

- It seems that the target of the regression stage is dynamic. I would encourage the authors to further analyze about the concerned instability.

- The confidence scores of the combined boxes are not introduced in method descriptions. It is important because score is essential for mAP. (good score also benifits mAP)

- Isn't the combined boxes the new boxes? Why the number of predictions remain? Does the mAP evaluator obtain more predictions than baseline?
> By design, our method yields the same number of predictions as current detectors and is compatible with any NMS variant.

- Is regression function r a learnable regression head? Why an additional regression refinement module is required? Isn't the combined boxes the prediction boxes? (We nevel refine NMS outputs with an additional regression module. The baseline method seems lack one refinement module)

- How does the detectors with other types of proposal (e.g. DETR with non-two-stage-initialized object queries, FCOS with point based dense head) be integrated seamlessly by UoI? It seems that this method can only be a box refinement module for these non-box-proposal methods.
> Our plug-and-play method integrates seamlessly into any detection architecture with minimal modifications

---

> ### Author Response · Authors · 2024-11-24
> **Response 1/2**
>
> We thank the reviewer for their appreciation and the opportunity to clarify our intuitions, methodological choices and expand our experimental justification.
>
> > **True alternative to original paradigm**
>
> To further validate the flexibility of our framework, we performed additional experiments showing that modifying individual components, such as the regression loss and the grouping strategy, still preserves the efficacy of our approach. For example, replacing the L1 loss with IoU-based losses GIoU [a], DIoU [b] and Alpha-IoU [c] enhances performance, as shown in the table below:
>
> | IoU Loss Type | Base ($mAP$) | Base w/ UoI ($mAP$) | Base ($AP_{75}$) | Base w/ UoI ($AP_{75}$) |
> |---------------|------------|--------------------|--------------|----------------------|
> | GIoU             | 38.0       | **38.6**   | 41.1         | **42.0**     |
> | DIoU            | 38.1       | **38.8**   | 41.1         | **41.9**     |
> | Alpha-IoU    | 38.9       | **39.4**   | 41.8         | **42.6**     |
>
> Similarly, our grouping mechanism remains effective across NMS variants like Cluster-NMS [d] and Soft-NMS [e], as demonstrated in the following table:
>
> | NMS Type | Base ($mAP$) | Base w/ UoI ($mAP$) | Base ($AP_{75}$) | Base w/ UoI ($AP_{75}$) |
> |---------------|------------|--------------------|--------------|----------------------|
> | NMS             | 37.4       | **38.1**   | 40.4         | **40.9**     |
> | Cluster-NMS             | 37.6       | **38.4**   | 40.4         | **41.0**     |
> | Soft-NMS      | 38.2       | **38.8**   | 40.9         | **41.7**     |
>
> We hope these results and analyses convincingly justify the viability and robustness of our approach as a true alternative to the conventional box refinement paradigm.
>
> [a] Rezatofighi H, et al. Generalized intersection over union: A metric and a loss for bounding box regression. CVPR 2019.
>
> [b] Zheng Z, et al. Distance-IoU loss: Faster and better learning for bounding box regression. AAAI 2020.
>
> [c] He J, et al. $\alpha $-IoU: A family of power intersection over union losses for bounding box regression. NeurIPS 2021.
>
> [d] Zheng Z et al. Enhancing Geometric Factors in Model Learning and Inference for Object Detection and Instance Segmentation. IEEE transactions on cybernetics, 2021
>
> [e] Bodla N, et al. Soft-NMS--improving object detection with one line of code. ICCV 2017
>
> ___
>
> > **Performance analysis on different object sizes**
>
> We thank the reviewer for sharing the insight. Our approach indeed profits from having multiple object proposals available. For the COCO dataset, the average number of proposals is 12, 8 and 3 for large, medium and small objects, respectively. Small objects typically have fewer proposals, and in cases with only one proposal, our method defaults to the baseline (see similar comment by Reviewer zZMj). To analyze this further, we grouped small objects based on the number of proposals they received (e.g., 1, 3, or 5 proposals) and calculated the corresponding AP scores for each group. The table below shows that AP_small improves as the number of proposals increases, highlighting that our method effectively utilizes multiple proposals to enhance performance for small objects. We will include a discussion on very small objects and the anticipated effects in the main paper.
>
> | Proposals per small object | $AP_{small}$ |
> |----------------------|----------------|
> | 1                    | 22.5           |
> | 3                    | 22.9           |
> | 5                    | 23.3           |
>
> ___
>
> > **Stability in light of dynamic target allocation**
>
> The reviewer correctly observes that our regression targets are dynamically assigned to parts of objects based on proposals, unlike the fixed object-level targets in the baseline. To analyze the stability of this assignment, we examined both training and testing stages. During training, our regression loss—associated with these dynamic targets—is approximately half of the baseline regression loss, indicating improved stability (as shown in Figure 3(b) of the main paper). During testing, Table 5 demonstrates that our method achieves a $10.7\%$  higher localization accuracy than the baseline ($64.4\%$ vs. $53.7\%$), ensuring proper localization capability to dynamic targets. We will update the text in these ablations in the main paper to explicitly reflect the stability of our dynamic target assignment, ensuring this concern is fully addressed.
>
> ___
>
> **<CONTINUED>**

---

> ### Author Response · Authors · 2024-11-24
> **Response 2/2**
>
> > **Confidence scores in methods description**
>
> The reviewer is correct that assigning new confidence scores might improve $mAP$. To evaluate this, we added a classification head for the combined boxes for the FRCNN baseline, resulting in marginal improvements in $mAP$ and $AP_{75}$, as shown in the table below. Since these gains are minimal and our primary improvements come from the localization, (Table 5 of the main paper), we chose to omit the additional classification stage in the main method. We will add the ablation to the supplemental.
>
> | Method                        | $mAP$  | $AP_{75}$ |
> |-------------------------------|-----|-------|
> | Original UoI                  |  38.1   |  40.9     |
> | UoI with 2nd classification stage |  **38.2**   |  **41.1**     |
>
> ___
>
> > **Same number of predictions as baselines**
>
> We confirm the total number of final predictions fed to the $mAP$ evaluator for the baseline and our approach are the same, as decided by the number of regions formed during the grouping stage.
>
> ___
>
> > **Extra regression head**
>
> Yes, the regression function $r$ is a learnable regression head. Our method includes an additional regression stage after grouping to refine the combined boxes into tighter-fitting bounding boxes. For baseline methods, adding a second regression head does not yield significant benefits, as both heads perform the same task—predicting the full object bounding box. In contrast, our approach assigns distinct roles: the first regression head focuses on parts of the object, while the second refines the union of these parts into an accurate bounding box. This makes the additional regression effective, as shown in the table below. We will add this ablation to the supplemental material.
>
> | Method               | $mAP$ | $AP_{75}$ |
> |--------------------------|--------|-----------|
> | FRCNN                     | 37.4 | 40.4   |
> | FRCNN w/ 2nd regression    | 37.4 | 40.5   |
> | FRCNN w/ UoI                      | **38.1** | **40.9**   |
>
> ___
>
> > **Adaptability to other methods**
>
> We clarify our method is not limited to traditional proposals and can be applied to methods with regions, grids, or queries, as demonstrated in the main paper (Tables 1 and  2). We have already shown its integration with grid-based YOLO and query-based Deformable DETR, highlighting its flexibility. The concept of "proposals" is used loosely to include regions and anchors, which are common across most detection frameworks, making our approach broadly applicable with minimal modifications.
>
> **Details of modifications:** For YOLO, we modify its object-to-grid assignment strategy by using an IoU-based criterion instead of the traditional center-based approach, assigning objects to multiple grid cells if they overlap sufficiently. Each anchor is tasked with regressing the part of the object it overlaps with best, and during inference, our Intersection-based Grouping merges these predictions. For Deformable DETR, we treat queries as proposals and redefine the matching process to focus on object parts, assigning queries to specific regions of the ground truth and combining them during inference.
>
> While most methods, including YOLO and Deformable DETR, can incorporate our Union-over-Intersection with minor modifications, unique architectures like FCOS, which rely on keypoints rather than rectangular anchors, may require additional adaptations. We will clarify the adaptability of our approach in the main paper.
>
> ___
>
> We hope our responses have clarified concerns regarding object sizes, dynamic regression targets, and integration with various detection architectures. Additionally, we have addressed aspects like confidence scoring and compatibility with non-box-proposal methods, demonstrating the flexibility and robustness of our plug-and-play design. We are happy to provide further clarifications if needed.

---

> ### Comment · Area_Chair_cV87 · 2024-11-25
>
> Dear Reviewer GWg2,
>
> Could you kindly review the rebuttal thoroughly and let us know whether the authors have adequately addressed the issues raised or if you have any further questions.
>
> Best,
>
> AC of Submission10907

---

> ### Comment · Reviewer_GWg2 · 2024-11-25
>
> Thanks for the feedback from the authors. I'm still concern on the adaptability.
>
> > For Deformable DETR, we treat queries as proposals and redefine the matching process to focus on object parts, assigning queries to specific regions of the ground truth and combining them during inference.
>
> So the query need to be box proposal? So this is DAB-Deformable-DETR, right?
>
> What about FCOS? I have not found the description in the revised manuscript.
>
>
> What I still care about is: if a method like Deformable DETR (require modifying query designing like DAB-DETR) and FCOS (point based) need so many designing modification, the description
>
> > Our plug-and-play method integrates seamlessly into **any detection architecture** with minimal modifications, significantly improving object localization and instance segmentation.
>
> is not sound.

---

> > ### Author Response · Authors · 2024-11-26
> >
> > > **Explanation of adaptability of our approach**
> >
> > The reviewer is right that our claim on compatibility for *any* detection architecture is inappropriate. Our framework integrates effectively with *most* detection architectures, including proposal-based, grid-based, and query-based methods, with minimal modifications. Architectures relying on keypoints instead of regions or boxes, like FCOS, may require additional adaptations. We will better highlight what architectures are currently (in)compatible and have softened our compatibility claim throughout the paper.
> >
> > We clarify that to adapt Deformable DETR for our Union-over-Intersections, it does not resemble DAB-DETR. DAB-DETR introduces dynamic anchor boxes to its queries, allowing them to adaptively refine their positions. In Deformable DETR, each query predicts a bounding box through the regression head after the decoder. We retain this Deformable DETR process for bounding box prediction. However, in standard Deformable DETR, these predicted boxes are matched to full ground truth boxes via Hungarian matching, assigning the best query to each ground truth box. In contrast, our approach breaks the ground truth box into four quadrants, assigning each query to a specific part of the ground truth rather than the whole. Then we do our grouping and merging. We have clarified our adaptation for Deformable DETR in the main paper. We hope we were able to address your concern regarding this.

---

> > > ### Comment · Reviewer_GWg2 · 2024-11-27
> > >
> > > I acknowledge and appreciate the authors’ efforts to address feedback by softening the overly broad claims regarding compatibility.
> > >
> > > I am willing to raise my rating.
> > >
> > > But I still need discuss with the authors and suggest to enhance the relevant sections.
> > >
> > > The description of the modifications to Deformable DETR is very brief and lacks detail and discussion.
> > >
> > > I'm trying to understand it.
> > >
> > > > In Deformable DETR, each query predicts a bounding box through the regression head after the decoder. Instead of matching these boxes to full ground truth boxes, we divide the ground truth into quadrants, assign queries to specific parts for part-based regression.
> > >
> > > It seems that the authors treat the single-stage, non-box-refinement Deformable DETR as a black box, enabling detectors based on set prediction (including Sparse RCNN, DETR, etc.) to support UoI post-processing during inference by predicting a quarter-sized box corresponding to one quadrant of the ground truth box. (Am i right? if not, please correct me.)
> > >
> > > I consider this is a clever and reasonable design. It could even be further extended (for instance, by predicting boxes smaller than a quarter that include the vertex but remain within the corresponding quadrant. I am not suggesting that the authors add additional experiments, as I believe the current experiments already demonstrate the effectiveness of the approach.). This seems reasonable because set prediction detectors are widely recognized for their ability to predict non-overlapping boxes inherently, and predictions within quadrants align seamlessly with this characteristic.
> > >
> > > I recommend that the authors provide a more detailed description and discussion in the supplementary materials about compatibility. As a reviewer of this paper, although I generally acknowledge this manner, I did not fully understand until I forced myself to think it through carefully. I think this lack of clarity could be avoided in the final manuscript.
> > >
> > > I also suggest avoiding the use of “most” (e.g., in abstract) and instead specifying which types of detection frameworks the method is applicable to. This would help users determine early on whether the approach could enhance their detector, saving time.

---

> > > > ### Author Response · Authors · 2024-11-27
> > > >
> > > > Thank you for upgrading your score and making further suggestions for improvement. We clarify in the abstract, introduction and related work of the updated manuscript that our method is beneficial for proposal-based, grid-based, and query-based object detectors.
> > > >
> > > > Regarding our adaptation of Deformable DETR, you are indeed correct in your interpretation—our method treats Deformable DETR as a black box. So it is adaptable to detectors based on set prediction (including Sparse RCNN, DETR, etc.). We predict a quarter-sized box corresponding to one quadrant of the ground truth box by modifying the target for regression during training. We are grateful for your insightful suggestions about extending the approach to predict even smaller boxes within quadrants. This is something we want to try in future  works. We have included a detailed discussion of the compatibility and adaptations for various architectures, including Deformable DETR and YOLO in the supplemental. Thanks again.

---

> > > > > ### Comment · Reviewer_GWg2 · 2024-11-28
> > > > >
> > > > > Thanks for the feedback, my questions have been resolved.

---

> > > > > > ### Author Response · Authors · 2024-12-01
> > > > > >
> > > > > > Thank you for engaging with us during the rebuttal and upgrading your score. Your comments made our paper stronger.

---

### Official Review · Reviewer_zZMj · 2024-11-08

**Soundness:** 4
**Presentation:** 4
**Contribution:** 4
**Rating:** 8
**Confidence:** 5

**Summary:**

This paper revisits the problem of predicting box locations in object detection architectures. Traditional approaches typically regress box proposals to maximize the intersection-over-union (IoU) score with the ground truth and then apply non-maximum suppression (NMS) to retain only the highest scoring box in each region. The authors argue that both steps are suboptimal: regressing proposals to the entire ground truth is a challenging task, and NMS ignores potentially useful information from other boxes. To address these issues, the authors propose a simpler method—regressing proposals only to the intersection area between the proposal and the ground truth, rather than the entire ground truth, thus avoiding the need for proposals to extrapolate beyond their visual scope and improving localization accuracy. Additionally, they suggest generating the final box outputs by taking the union of the regressed intersections from all boxes in a region, rather than using a winner-takes-all approach, thus preserving valuable information from other boxes. This plug-and-play method integrates seamlessly into any detection architecture with minimal modifications and significantly enhances object localization and instance segmentation. The experimental results demonstrate its broad applicability and impressive performance across a variety of detection and segmentation tasks.

**Strengths:**

The "UoI" strategy proposed in this paper is highly innovative and practical, as it can seamlessly integrate into existing two-stage object detection networks, significantly improving detection performance. The authors creatively introduce the idea of regressing proposals to the intersection area with the ground truth, rather than the entire ground truth, addressing the challenges of regressing to the full target region. This not only improves localization accuracy but also offers an elegant and practical solution that can be easily adopted within current detection architectures. The paper is clearly written, with a logical structure that makes the proposed UoI method easy to understand. The figures and pseudocode are well-presented, offering an intuitive explanation of the approach. Furthermore, the extensive ablation experiments validate the effectiveness of UoI across various detectors, further strengthening the paper’s contribution. Overall, the paper excels in originality, clarity, and practical applicability.

**Weaknesses:**

While the paper excels in terms of methodological innovation, clarity of writing, and the presentation of figures and pseudocode, there is room for improvement in the design of comparative and ablation experiments. Specifically, the validation of the proposed UoI strategy is limited to a few methods, which may constrain its generalizability. For example, Table 1 only compares Faster R-CNN and Def-DETR’s detection performance on the PASCAL VOC dataset, while Table 2 provides comparisons for five methods (Faster R-CNN, Mask R-CNN, Cascade R-CNN, YOLOv3, and Def-DETR) on the MS-COCO dataset. However, why are methods like Mask R-CNN, Cascade R-CNN, and YOLOv3 not included in the PASCAL VOC comparison? Including these methods would strengthen the results and provide a more comprehensive evaluation.

Moreover, Tables 4-7 and Figure 3 mainly show results for Faster R-CNN, without testing the UoI strategy on other classic detection architectures. To further validate the effectiveness of UoI, it would be beneficial to conduct experiments on additional well-established methods, demonstrating its applicability and advantages across a broader range of architectures. This would enhance the paper’s persuasiveness and provide stronger evidence for the practical utility of the proposed approach.

**Questions:**

1.	To enhance the overall aesthetic consistency of the paper's layout, a few minor formatting adjustments are recommended:

1）	Some bolded paragraph titles (e.g., the bold text followed by a period, such as "Problem Statement." in line 150) are set on a new line rather than immediately followed by the paragraph content. This may be intentional due to title length, or it might be a formatting issue. It is suggested to align these paragraph titles directly with the content following them. For example, "Intersection-based Grouping." in line 203 and "Regressing to intersections is simply an easier task." in line 448 should follow this format.

2）	The title "Inference cost" in line 267 lacks a period and should be updated to "Inference Cost." Additionally, the caption of Figure 4 ("Qualitative results…") is not bolded like the caption in Figure 5; please consider ensuring consistency with Figure 5's formatting.

3）	The paragraph titles throughout the paper vary between Title Case and Sentence Case. Although I will not list each instance here, please consider standardizing these for consistency.

2.	To ensure accuracy and clarity, a few content-related questions have also arisen:

1）	In line 220, the text states, "The regression function r : B → R maps each combined box to a final target box." Here, does R refer to the set of all ground truths? Previous sections define G or T as representing all ground truths, so it is unclear why R is used here instead of G or T. Could this be clarified?

2）	To guarantee the precision of the pseudocode in the Post-Processing Stage on the right side of Figure 2, could you confirm if it would be necessary to include "P ← P \ p_i" in the conditional statement within the red dashed box to ensure the "while P ≠ empty do" loop functions correctly (assuming the pseudocode within the green solid box is disregarded)?

3.	Could you clarify why the conditional statement in the pseudocode within the green solid box on the right side of Figure 2 uses "iou(H, pi) ≥ k" rather than "iou(M, bi) ≥ k" as in the red dashed box? Has there been any experimentation to validate the effectiveness of "iou(H, pi) ≥ k" over "iou(M, bi) ≥ k"? If not, please consider including such justification.

4.	In Table 3, there is a Comparison of instance segmentation on MS-COCO data, showing considerable improvements in AP for various object sizes (i.e., small, medium, and large objects as defined by COCO). The COCO definition of "small objects" encompasses those with an area under 32×32 pixels. Compared with small-object detection tasks in infrared imaging (where small objects typically occupy less than 10×10 pixels and may even be as small as a few pixels), COCO's small objects are relatively large. Could the authors clarify whether their proposed UoI method would also be effective for these considerably smaller targets?

5.	Figure 3(c) demonstrates that on the MS-COCO dataset, Faster R-CNN achieves optimal performance with the UoI strategy when the intermediate group size is 5. Have the authors considered studying the effect of varying the IoU threshold k in the UoI strategy (i.e., the IoU threshold used in the conditional statement in the pseudocode within the green solid box on the right side of Figure 2)?

6.	Figures 4 and 5 highlight certain limitations of the proposed UoI approach, specifically in handling crowded scenes. Could the authors specify if these crowded scenes refer to instances of multiple objects or instances of the same class, or do they also include multiple instances of different classes? Additionally, might the size of objects or instances in these crowded scenes significantly restrict UoI’s performance? Please provide a detailed discussion on these aspects.

---

> ### Author Response · Authors · 2024-11-24
> **Response 1/2**
>
> We thank the reviewer for their encouragement and the suggestions for further improvement.
>
> > **Additional comparisons per table**
>
> Following the reviewer suggestion, we present an updated version of Table 1, which now includes results for Mask R-CNN, Cascade R-CNN, and YOLOv3 on the PASCAL VOC dataset, in addition to Faster R-CNN and Deformable DETR. In all cases, adding our Union-over-Intersections improves detection results, with stronger improvements for higher overlap thresholds. We will put this table in the supplemental material.
>
> | Method           | AP_50 | AP_70 | AP_90 |
> |-------------------|-------|-------|-------|
> | Faster R-CNN     | 71.2  | 51.3  | 6.8  |
> | Faster R-CNN w/ UoI    | 72.9  | 55.3  | 11.3  |
> | Mask R-CNN       | 73.1  | 52.7  | 7.9  |
> | Mask R-CNN w/ UoI      | 74.0  | 56.9  | 13.1  |
> | Cascade R-CNN    | 77.5  | 54.1  | 10.5  |
> | Cascade R-CNN w/ UoI   | 78.8  | 57.9  | 14.8  |
> | YOLOv3           | 56.1  | 41.7  | 5.3  |
> | YOLOv3 w/ UoI          | 57.0  | 44.2  | 8.8  |
> | Deformable DETR  | 83.2  | 68.5  | 22.1  |
> | Deformable DETR w/ UoI | 84.1  | 70.3  | 26.5  |
>
> ___
>
> > **Additional architectures for ablations**
>
> As requested, we provide additional results for the localization effectiveness of UoI on Faster R-CNN and Cascade R-CNN. The first table evaluates classification accuracy and localization mIoU (Table 5 of main paper), while the second examines LRP metrics (Table 6 of main paper), demonstrating that main improvement for results with the proposed Union-over-Intersections (UoI) indeed comes from localization, irrespective of the base architecture.
>
> | Method                   | Classification Accuracy [$\%$] | Localization mIoU [$\%$] |
> |--------------------------|-----------------------------|------------------------|
> | Faster R-CNN | 76.4                       | 53.7                  |
> | Faster R-CNN w/ UoI            | 76.5                       | 64.4                  |
> | Cascade R-CNN | 78.6                       | 56.2                  |
> | Cascade R-CNN w/ UoI            | 78.8                       | 68.3                  |
>
> | Method        | LRP (Error) | $LRP_{Loc}$ | $LRP_{FP}$ | $LRP_{FN}$ |
> |--------------------|------------------|-------------|------------|------------|
> | Faster R-CNN         | 67.6            | 17.2        | 24.2       | 44.3       |
> | Faster R-CNN w/ UoI  | 65.3            | 12.7        | 23.9       | 43.8       |
> | Cascade R-CNN         | 64.4            | 15.2        | 24.0       | 41.7       |
> | Cascade R-CNN w/ UoI  | 62.8            | 10.3        | 23.6       | 40.9       |
>
> We are running additional experiments to validate the UoI strategy on YOLO v3 and  Deformable DETR for all datasets for results on Tables 4-7 and will include these results in the supplemental material.
>
> ___
>
> > **Aesthetic consistency of the paper**
>
> Thank you for highlighting the formatting inconsistencies. We have aligned paragraph titles with their content, standardized title casing, and ensured consistent formatting across figures and captions in the updated version of the paper.
>
> ___
>
> > **Content related queries**
>
> 1. By the regression function $r: B \to R$, we mean that the combined boxes $B$ are refined into final regressed boxes $R$, which are compared to the ground truth boxes, denoted as $G$ or $T$, using the loss function: $L_{\text{refinement}} = \sum_{j=1}^{m} \sum_{i \in \{x, y, w, h\}} |r(B_{ji}) - T_{ji}|$. In this context, $R$ is not intended to represent the ground truth but rather the set of final predictions obtained after refinement. We will revise the wording in the main manuscript for improved clarity.
>
> 2. We confirm that it is not necessary to include "P ← P \ p_i" in the conditional statement within the red dashed box. The red dashed box represents standard NMS, which groups based on the predicted bounding boxes $B$, not the proposals $P$. Our approach instead uses the proposals $P$—the boxes before regression—for grouping. Since the red dashed box exclusively operates on $B$, including $P$ there is unnecessary.
>
> ___
>
> > **Conditional statement in pseudocode**
>
> The use of "iou(H, p_i) >= k" in the green solid box reflects a key distinction in our method: while standard NMS does grouping based on regressed boxes $B$, our approach uses original proposals $P$ for grouping. This is because our regression stage outputs intersections, not full bounding boxes. Grouping by IoU between intersections $I$ (as in "iou(M, b_i) >= k") is ineffective, as intersections often lack sufficient overlap. We tested this alternative and confirmed it performed poorly: grouping by regressed intersections $I$ resulted in 33.6 $mAP$, while grouping by proposals $P$ achieved 38.1 $mAP$. Grouping by proposals $P$, which better represent object regions, is more effective for our part-based approach. We will clarify this in the main paper.
>
> ___
>
> **<CONTINUED>**

---

> ### Author Response · Authors · 2024-11-24
> **Response 2/2**
>
> > **Effects on tiny objects**
>
> Thank you for sharing the insight on very tiny objects. Indeed, our approach profits from having multiple object proposals available. For the COCO dataset, the average number of proposals is 12, 8 and 3 for large, medium and small objects respectively.
>
> Small objects typically have fewer proposals, and in cases with only one proposal, our method defaults to the baseline (see similar comment by Reviewer GWg2). On COCO the small objects still have an average of three proposals, enabling our Union-over-Intersection to unify their information and contribute to the reported improvements. For even smaller objects (<10×10 pixels), the method's performance will depend on the base network and the number of intersecting proposals. Our approach is never worse than the baseline and will demonstrate an improvement when there is more than 1 proposal available. We will include a discussion on very small objects and the anticipated effect on our approach in the main paper.
>
> ___
>
> > **Effect of varying IoU threshold**
>
> We thank the reviewer for this observation. The number of proposals is influenced by the IoU threshold (k) used for grouping in the UoI strategy. We set this threshold to 0.5, consistent with standard NMS. Lowering it (e.g., 0.3) increases group sizes, while raising it (e.g., 0.7) reduces them. As shown in the table below for FRCNN with a Resnet-101 backbone, the optimal performance (mAP) shifts to 4 proposals for a 0.7 threshold but remains at 5 for a 0.3 threshold. However, optimal mAP values for both 0.3 and 0.7 are lower than 0.5, reaffirming the effectiveness of the default setting. We will add this ablation to the supplemental material.
>
> | IoU threshold k | 3 Proposals | 4 Proposals | 5 Proposals | 6 Proposals | 7 Proposals |
> |-----------|-------------|-------------|-------------|-------------|-------------|
> | 0.3       | 38.1        | 39.0        | 39.5        | 39.2        | 38.4        |
> | 0.5       | 39.3        | 39.8        | 40.3    | 39.2        | 38.5        |
> | 0.7       | 38.5        | 39.9    | 39.4        | 38.1        | 37.7        |
>
> ___
>
> > **Crowded Scenes impact**
>
> We thank the reviewer for this insightful observation. Our limitation in crowded scenes specifically concerns multiple instances of the same class. Instances of different classes are naturally separated during grouping. However, for same-class objects positioned closely, candidate boxes may merge into a single large box, as shown in Figure 4's "3 birds" example.
>
> Regarding object size, grouping is scale-invariant as it relies solely on IoU overlap. The challenge arises in the refinement stage after grouping, where regressing to the nearest object performs better for distinct, larger objects. Hence, this limitation is most evident with small, closely spaced objects of the same class, while larger or less crowded objects are less affected. We will add this discussion to the main paper.
>
> ___
>
> We hope our responses adequately address the concerns raised, particularly around the generalization of our method across datasets, architectures, and object sizes, as well as its robustness in crowded scenes and various IoU threshold settings. We are happy to provide further clarifications if needed.

---

> > ### Author Response · Authors · 2024-11-27
> >
> > Dear reviewer,
> >
> > As the discussion stage is ending soon, we wonder if our response answers your questions and addresses your concerns? Thanks again for your very constructive and insightful feedback!

---

### Author Response · Authors · 2024-11-24

We sincerely thank all reviewers for their constructive feedback and insightful suggestions, which greatly helped us refine and strengthen our paper. We believe to have addressed all concerns raised and provided the suggested additional ablations and experiments, to validate the robustness, flexibility, and applicability of our Union-over-Intersections method.

To ensure clarity and completeness, we have made modifications to the main paper to directly reflect the changes and created a detailed supplemental document with all new results and ablations. We hope our responses and updates address your concerns and demonstrate the efficacy of our approach across various scenarios.

We remain open to further clarifications or discussions and thank you once again for your valuable time and effort in reviewing our work.

---

### Meta-Review · Area_Chair_cV87 · 2024-12-17

**Metareview:**

(a) The authors propose the Union-over-Intersections (UoI) method, which improves object detection by focusing regression on intersection areas for better localization and combining proposal information through union in post-processing.

(b) The strengths of the paper include a simple, clear, and easily reproducible method, a well-structured and well-written presentation, sufficient analyses to demonstrate effectiveness, and a promising advancement in the field of object detection.

(c) The major weaknesses are the limited novelty of the method, as intersections are already addressed by IoU and IoU loss, and the Union of Proposals resembles established voting mechanisms. Additionally, the method's reliance on proposals raises concerns about generalizability to proposal-free approaches like FCOS.

(d) The most important reasons for acceptance are that this work changes the original detection pipeline, which regresses the full bounding box and uses the winner-takes-all NMS paradigm, to one that regresses parts of the box and merges proposals to obtain the final result. This is a very novel approach in the field of object detection, and the authors have demonstrated its effectiveness.

**Additional Comments On Reviewer Discussion:**

(a) Reviewer zZMj shows that the paper's weaknesses include limited comparative and ablation experiments, with the UoI strategy tested on only a few methods and datasets. Expanding the comparisons to include more detection architectures would strengthen the evaluation and demonstrate the broader applicability of the proposed approach. The authors successfully address the above issues.

(b) Reviewer GWg2 finds that the method may favor larger objects due to the "combine box" approach, potentially limiting its effectiveness for small objects. The dynamic target of the regression stage raises concerns about instability, and further analysis is needed. Additionally, the absence of confidence scores for combined boxes, the unchanged number of predictions, and the unclear role of the regression refinement module are all areas that need clarification. The first round of rebuttal addresses most of the concerns, the reviewer further shows concerns on the adaptability. The authors admit the limitation and clarify that they will better highlight what architectures are currently (in)compatible and have softened our compatibility claim throughout the paper.

(c) Reviewer fZZc concerns about the limited novelty of the method, as intersections are already addressed by IoU and IoU loss, and the Union of Proposals resembles established voting mechanisms. He shares similar concerns with Reviewer GWg2 regarding adaptability. The first round of rebuttal addresses most concerns, and the reviewer expresses interest in comparing the performance gains between voting and the proposed method through several rounds of discussion. The authors further demonstrate that their approach is robust to various test-time augmentations. The reviewer is satisfied and comments that he will raise the score to 6.

(d) Reviewer JFjT finds that the paper lacks comparison with IoU-based loss functions, making the method's superiority less convincing. Only Soft-NMS is tested, despite claims of compatibility with other NMS variants, suggesting the need for more ablation studies. Additionally, the method’s sensitivity to the number of proposals may limit its robustness across different detectors or datasets. The authors successfully address the above issues. The reviewer provides several suggestions to adjust or add content between the main paper, rebuttal, and supplementary material, and increases the score to 8.

---

### Decision · Program_Chairs · 2025-01-22

Accept (Spotlight)